# MCPLANNER: MULTI-SCALE CONSISTENCY PLANNING FOR OFFLINE REINFORCEMENT LEARNING

## ABSTRACT

Planning for long-horizon tasks is a significant challenge, often addressed with complex hierarchical methods that rely on multiple, independently trained models. These hierarchical approaches can be brittle and incur coherence issues. In this work, we introduce **M**ulti-scale **C**onsistency **Planner** (**MCPlanner**), a novel framework that leverages the unique properties of Generalized Consistency Trajectory Models (GCTMs) to create a fluid and unified planning hierarchy. Unlike prior generative models which are limited to mappings from noise to data, GCTMs can learn a direct, fully-traversable trajectory path between arbitrary data distributions. This crucial capability allows MCPlanner to unify high-level and low-level planning within a single model. Instead of training separate high-level and low-level planners, MCPlanner employs a single GCTM trained on end-to-end expert trajectories. At inference time, a seamless hierarchy emerges: coarse, long-horizon plans are generated by querying the model at a sparse temporal resolution, while dense, fine-grained motions are synthesized by querying the same model on the continuous path between these coarse waypoints. Our approach obviates the need for discrete hierarchical structures, offering a more elegant, efficient, and controllable solution to long-horizon planning. Furthermore, our experimental results demonstrate that MCPlanner achieves state-of-the-art performance across 35 challenging tasks on OGBench benchmark, by consistently outperforming prior approaches.

## 1 INTRODUCTION

Effectively planning over long horizons is a fundamental challenge in sequential decision-making, crucial for solving complex tasks that require reasoning over thousands of steps. Traditional approaches often struggle with the combinatorial complexity of searching vast state-action spaces, leading to computational intractability and suboptimal solutions. A dominant paradigm to tackle this complexity is *hierarchical planning* (Sacerdoti, 1974; Kaelbling & Lozano-Pérez, 2011), which decomposes a difficult problem into a series of more manageable subgoals. This allows for both high-level strategic reasoning, where abstract goals are formulated and sequenced, and low-level action generation, where precise movements are executed to achieve immediate objectives. However, this decomposition typically introduces challenges in maintaining inter-level coherence and managing the increased complexity of training and coordinating multiple independent models.

The recent success of generative models, particularly denoising diffusion models (Sohl-Dickstein et al., 2015; Ho et al., 2020), has led to their application in planning tasks. Prior works like Diffuser (Janner et al., 2022) leverages diffusion models with classifier guidance (Nichol et al., 2022), and Decision Diffuser (Ajay et al., 2023) employs classifier-free guidance (Ho & Salimans, 2022) by directly conditioning on returns during sampling. While effective for short-horizon tasks, these methods can be computationally expensive and sample inefficient. Subsequent works have extended these to long-horizon problems, often adopting hierarchical approaches that typically employ two separate diffusion models: a high-level planner for subgoal generation and a low-level planner for trajectory synthesis between them (Li et al., 2023; Chen et al., 2024; Hao et al., 2025). While powerful, this two-model paradigm exacerbates existing challenges. Separately trained planners often suffer from a lack of coherence, where the low-level model may struggle to execute subgoals proposed by the high-level one. Furthermore, this approach doubles the training and maintenance burden and can be computationally inefficient. This leads to a fundamental question: ***Is a rigid, two-model hierarchy the most effective and elegant way to solve long-horizon tasks?***

Figure 1: **Overview of MCPlanner: Previous Methods** often rely on separate high-level and low-level planners, which can lead to a **lack of coherence and consistency**. The high-level plan (left) defines abstract goals, while the low-level plan (middle) executes detailed actions, sometimes failing to align with the high-level intent (e.g., the warning sign). In contrast, **MCPlanner** introduces a **unified hierarchy** (right) where a single model possesses both coarse and fine planning abilities, ensuring seamless and coherent trajectory generation from abstract goals to precise actions.

In response to these limitations, we propose a powerful approach that utilizes a single, unified model to seamlessly transition between high-level, coarse planning and low-level, fine-grained control within the offline RL context. Such a unified model inherently eliminates the coherence problem prevalent in multi-model hierarchies and offers a more streamlined and efficient framework.

To achieve this, we turn to a new class of generative models that addresses the limitations of diffusion. Consistency Models (CMs) (Song et al., 2023) were introduced to overcome the slow sampling speed of diffusion models by learning a direct one-step mapping from any noisy sample to a clean data point. While offering significant speed-up, CMs are limited to learning only the endpoint of the generation process. Consistency Trajectory Models (CTMs) (Kim et al., 2024) extended this by learning the entire ordinary differential equation (ODE) trajectory, allowing for traversal between any two points in the generation path. However, both CMs and CTMs remain fundamentally tied to the diffusion framework, learning a path exclusively from Gaussian noise to data.

The key innovation we leverage is the Generalized Consistency Trajectory Model (GCTM) (Kim et al., 2025), which breaks this final limitation by integrating flow-matching (Lipman et al., 2022). GCTMs can learn a direct, fully-traversable ODE path between *two arbitrary data distributions*. This is precisely the property required for a unified planner: instead of mapping from noise to a trajectory, a GCTM can learn to map directly from the distribution of start states to the distribution of complete, successful trajectories. This ability to learn a direct, yet structured and multi-scale, mapping between two complex, meaningful distributions is the cornerstone of our approach.

In this paper, we introduce **M**ulti-scale **C**onsistency **P**lanner (**MCPlanner**), a unified planning framework that replaces rigid two-model hierarchies with a single GCTM trained end-to-end on full trajectories. At test time, a seamless hierarchy emerges by querying the same model at multiple temporal resolutions: sparse queries produce coherent high-level subgoals, while dense queries synthesize fine-grained motion between them. To ensure coherence and efficiency, MCPlanner (i) enforces a multi-scale consistency objective that promotes compositional consistency across temporal jumps, (ii) straightens the learned flow via conditional optimal-transport couplings for faster, more stable integration, and (iii) enables controllable and sample-efficient planning through a structured latent space with lightweight latent refinement. This unified design not only eliminates the coherence problem but also reduces training and inference costs, ultimately achieving state-of-the-art performance on OGBench tasks.

In summary, our contributions are as follows:

- We propose MCPlanner, a novel, unified hierarchical planner that uses a single generative model for both high-level subgoal generation and low-level trajectory synthesis.

- We introduce and enforce a multi-scale consistency objective that explicitly promotes compositional coherence across different temporal resolutions in the unified planning hierarchy.

- We enhance the learning of the GCTM flow through conditional optimal-transport couplings, leading to straightened ODE trajectories for faster, more stable integration and improved sample efficiency.

- We demonstrate that the latent space of our unified planner can be used to exert high-level, strategic control over the generated plans, allowing for dynamic adaptation of planning behavior.

- We evaluate MCPlanner on a wide variety of locomotion and manipulation tasks from OGBench benchmark and show that our method outperforms prior works by a wide margin.

## 2 RELATED WORKS

**Generative Models.** Generative models, particularly score-based and diffusion models (Sohl-Dickstein et al., 2015; Ho et al., 2020; Song et al., 2021; Karras et al., 2022), have emerged as a dominant force in machine learning, revolutionizing fields from image and text synthesis to drug discovery (Rombach et al., 2022; Podell et al., 2024; Nichol et al., 2022; Li et al., 2022; Gupta et al., 2024; Avdeyev et al., 2023). Their strength lies in their ability to learn complex, high-dimensional data distributions and generate high-fidelity samples. Initially popularized in computer vision, these models have seen rapid adoption and innovation. A significant area of research has focused on accelerating the iterative sampling process, which is notoriously slow. This has led to the development of techniques like distillation (Luhman & Luhman, 2021; Salimans & Ho, 2022; Meng et al., 2023; Berthelot et al., 2023; Shao et al., 2023; Wang et al., 2025). However, these distillation models still experience slow convergence or extended runtime. Consistency models (CMs) (Song et al., 2023; Song & Dhariwal, 2024; Geng et al., 2025; Wang et al., 2024a; Lee et al., 2025; Lu & Song, 2025) are a new type of generative models that support fast and high-quality generation. They do not rely on a pretrained diffusion model to generate training targets but instead leverage an unbiased score estimator. Consistency trajectory models (CTMs) (Kim et al., 2024) generalize consistency models by enabling the prediction between any two points on the same ODE trajectory. Their training objective becomes more challenging than standard consistency models that only care about the mapping from intermediate points to the data endpoints. Generalized consistency trajectory models (GCTMs) (Kim et al., 2025) extend CTMs by enabling one-step translation between arbitrary distributions, surpassing the limitations of traditional CTMs confined to Gaussian noise to data transformations.

**Generative Models for Planning.** Many works have studied the applications of generative models, particularly denoising diffusion models (Ho et al., 2020), for planning (Janner et al., 2022; Ajay et al., 2023; Pearce et al., 2023; Wang et al., 2023; Lu et al., 2025; Zhu et al., 2024). Diffusion-based planning has been widely adopted into various fields, such as autonomous driving (Liao et al., 2025; Yang et al., 2024; Wang et al., 2024b), task planning (Yang et al., 2023; Fang et al., 2024) and motion planning (Carvalho et al., 2023; Luo et al., 2024). Recently, they have also been extended to hierarchical planning to tackle long-horizon tasks Li et al. (2023); Chen et al. (2024); Hao et al. (2025); Ma et al. (2024), however they employ two separate diffusion planners making them incoherent. In contrast, our method uses a single model to generate subgoals as well as the dense trajectory.

## 3 PRELIMINARIES

**Problem Formulation.** We formalize the long-horizon planning problem within the framework of a controlled Markov process (a Markov Decision Process (MDP) without rewards), defined by the tuple $M = \{S, A, P, \gamma, d_0\}$, where $S$ is the state space, $A$ is the action space, $P : S \times A \to S$ is the state transition function, $\gamma \in [0, 1)$ is the discount factor, and $d_0$ is the initial state distribution.

Our goal is to learn a planner that, given an initial state $s_{start} \in S$ and a goal state $s_{goal} \in S$, can generate a full trajectory of state-action pairs, $\tau = ((s_0, a_0), (s_1, a_1), \ldots, (s_T, a_T))$, such that $s_0 = s_{start}$ and $s_T = s_{goal}$. We operate in an offline setting, where we have access to a fixed dataset $\mathcal{D}$ of expert trajectories. Our task is to learn a generative model $p(\tau | s_{start}, s_{goal})$ that can produce novel, successful trajectories for previously unseen start-goal pairs. By generating actions directly, our planner obviates the need for a separate inverse dynamics model.

**Diffusion Models.** Denoising diffusion models are powerful generative models that learn a data distribution $p(\mathbf{x})$ by reversing a predefined noising process. The process starts with a clean data sample $\mathbf{x}_0$ and gradually adds Gaussian noise over a sequence of $T$ timesteps. The forward noising process is defined as:

$$q(\mathbf{x}_t | \mathbf{x}_{t-1}) = \mathcal{N}(\mathbf{x}_t; \sqrt{1 - \beta_t}\mathbf{x}_{t-1}, \beta_t \mathbf{I}) \tag{1}$$

where $\{\beta_t\}_{t=1}^T$ is a fixed variance schedule. A key property is that we can sample a noisy version of $\mathbf{x}_0$ at any timestep $t$ in a single step:

$$q(\mathbf{x}_t|\mathbf{x}_0) = \mathcal{N}(\mathbf{x}_t; \sqrt{\bar{\alpha}_t}\mathbf{x}_0, (1 - \bar{\alpha}_t)\mathbf{I}) \tag{2}$$

where $\alpha_t = 1 - \beta_t$ and $\bar{\alpha}_t = \prod_{i=1}^t \alpha_i$.

The model learns to reverse this process. A neural network, $\epsilon_\theta(\mathbf{x}_t, t)$, is trained to predict the noise that was added to create $\mathbf{x}_t$ from $\mathbf{x}_0$. The training objective is typically a simplified mean-squared error loss:

$$\mathcal{L}_{\text{diff}} = \mathbb{E}_{t,\mathbf{x}_0,\epsilon} \left[ ||\epsilon - \epsilon_\theta(\sqrt{\bar{\alpha}_t}\mathbf{x}_0 + \sqrt{1 - \bar{\alpha}_t}\epsilon, t)||^2 \right] \tag{3}$$

At inference time, a sample is generated by starting with pure noise $\mathbf{x}_T \sim \mathcal{N}(0, \mathbf{I})$ and iteratively applying the learned denoising function to step backward in time until a clean sample $\mathbf{x}_0$ is produced. While effective, this iterative process requires many steps, making it computationally slow.

**Consistency Models (CMs).** (Song et al., 2023) were introduced to address the slow sampling speed of diffusion models. The core idea is to learn a function that can map any noisy sample directly back to the clean sample in a single step. CMs are based on the probability flow (PF) ODE of the diffusion process, where all points on the same ODE trajectory correspond to the same starting point $\mathbf{x}_0$. A consistency function, $f_\theta(\mathbf{x}_t, t)$, is trained to embody this property:

$$f_\theta(\mathbf{x}_t, t) \approx \mathbf{x}_0 \tag{4}$$

This allows for one-step generation, $\mathbf{x}_0 \approx f_\theta(\mathbf{x}_T, T)$, but this single-step mapping is restrictive.

**Consistency Trajectory Models (CTMs)** (Kim et al., 2024) extended this idea by learning not just the endpoint, but the entire trajectory. A CTM learns the integral of the PF-ODE, which describes the path of a sample from noise to data. This integral is denoted $G(\mathbf{x}_t, t, s)$, which transports a sample $\mathbf{x}_t$ at time $t$ to its position on the same trajectory at time $s$. A CTM parameterizes this solution as:

$$G_\theta(\mathbf{x}_t, t, s) = \frac{s}{t}\mathbf{x}_t + \left(1 - \frac{s}{t}\right) g_\theta(\mathbf{x}_t, t, s) \tag{5}$$

where $g_\theta$ is a neural network. The model is trained by minimizing two key losses. The first is a distillation loss, $\mathcal{L}_{CTM}$, which enforces self-consistency. It ensures that a one-step jump from time $t$ to $s$ is close to a two-step jump (from $t$ to an intermediate time $u$, then to $s$):

$$\mathcal{L}_{\text{CTM}}(\theta) = \mathbb{E}_{t,s,u,\mathbf{x}_0} \left[ d\left(G_\theta(\mathbf{x}_t, t, s), G_{sg(\theta)}(\mathbf{x}_{t \to u}, u, s)\right) \right] \tag{6}$$

where $sg$ is the stop-gradient operator. The second is the Denoising Score Matching (DSM) loss, $\mathcal{L}_{DSM}$, which anchors the model to the ground truth data by training $g_\theta$ to be an effective denoiser:

$$\mathcal{L}_{\text{DSM}}(\theta) = \mathbb{E}_{t,\mathbf{x}_0,\epsilon} \left[ ||\mathbf{x}_0 - g_\theta(\mathbf{x}_t, t, t)||_2^2 \right] \tag{7}$$

However, CTMs are still tied to the original diffusion formulation, learning the path only from Gaussian noise to data.

**Generalized Consistency Trajectory Models (GCTMs)** (Kim et al., 2025), which our work is built on, break this final limitation. GCTMs use Flow Matching to learn an ODE path between *two arbitrary data distributions*, $q(\mathbf{x}_1)$ and $q(\mathbf{x}_0)$. The learned ODE is given by:

$$d\mathbf{x}_t = t^{-1}(\mathbf{x}_t - \mathbb{E}_{q(\mathbf{x}_0|\mathbf{x}_t)}[\mathbf{x}_0]) \, dt \tag{8}$$

A GCTM learns the solution to this more general ODE, using the same parameterization $G_\theta(\mathbf{x}_t, t, s)$ from Eq. 5. This allows the model to transport a sample from a start distribution $q(\mathbf{x}_1)$ (at $t = 1$) to a goal distribution $q(\mathbf{x}_0)$ (at $t = 0$). The training objective mirrors that of CTMs, but the losses are generalized. The consistency loss, $\mathcal{L}_{GCTM}$, has the same form but operates on the new ODE. The DSM loss is replaced by the Flow Matching (FM) loss, $\mathcal{L}_{FM}$, which serves the same purpose of anchoring the model to the target distribution $q(\mathbf{x}_0)$:

$$\mathcal{L}_{GCTM}(\theta) = \mathbb{E}_{t,s,u,(\mathbf{x}_0,\mathbf{x}_1)\sim q} \left[ d\left(G_\theta(\mathbf{x}_t, t, s), G_{sg(\theta)}(\mathbf{x}_{t \to u}, u, s)\right) \right] \tag{9}$$

$$\mathcal{L}_{FM}(\theta) = \mathbb{E}_{t,(\mathbf{x}_0,\mathbf{x}_1)\sim q} \left[ ||\mathbf{x}_0 - g_\theta(\mathbf{x}_t, t, t)||_2^2 \right] \tag{10}$$

This ability to learn a direct, yet fully traversable, mapping between two complex distributions is the key property we exploit in our planning framework.

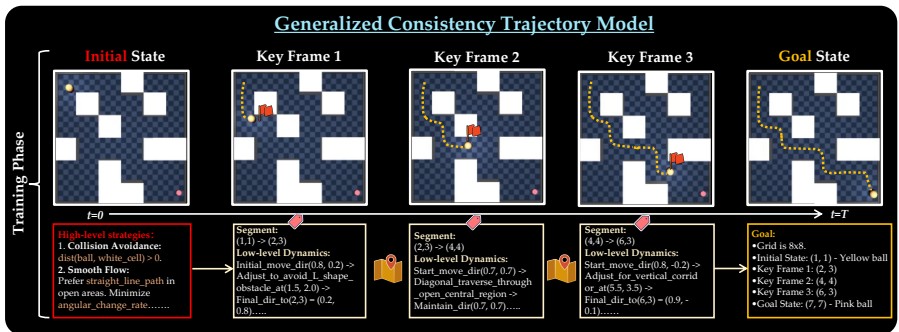

Figure 2: **MCPlanner.** Our framework leverages a single **Generalized Consistency Trajectory Model (GCTM)** to learn multi-scale planning. The GCTM is trained on end-to-end expert trajectories, learning both **high-level strategies** (e.g., collision avoidance, smooth flow) and intricate **low-level dynamics** for trajectory segments. At each segment, the model learns fine-grained motion synthesis, obviating the need for separate high-level and low-level planners and ensuring a unified and coherent planning hierarchy.

## 4 METHOD

This paper explores the integration of GCTMs into a novel planning architecture for offline RL. In the following, we discuss how we use a single, unified GCTM for a multi-scale trajectory optimization process.

### 4.1 TRAINING PROCESS

Our MCPlanner employs a single Generalized Consistency Trajectory Model (GCTM), $G_\theta$, trained on a dataset of full, end-to-end expert trajectories. This GCTM learns to map from a *start-goal* condition to the distribution of complete trajectories. We represent a full expert trajectory as a sequence of state-action pairs, $\tau = ((s_0, a_0), (s_1, a_1), \ldots, (s_T, a_T))$. Given a start-goal condition $c = (s_{start}, s_{goal})$, we set $\mathbf{x}_0 \equiv \tau$ and deterministically construct $\mathbf{x}_1$ by linearly interpolating states from $s_{start}$ to $s_{goal}$ across the trajectory horizon and setting actions to zero. This formulation yields convex interpolants $(1 - t)\mathbf{x}_0 + t\mathbf{x}_1$ that define the training path, obviating the need for any learnable encoder and establishing a direct connection between the condition and the trajectory generation process.

**Flow Straightening via Conditional Optimal Transport.** To further enhance sample and compute efficiency by reducing the curvature of the Flow Matching (FM) ODE, we employ an entropy-regularized optimal-transport (OT) coupling between batches of expert trajectories and their corresponding deterministic trajectory priors. This coupling introduces an inductive bias, encouraging the start and end distributions of the learned flow to be close in Euclidean distance, thereby yielding straighter ODE trajectories. This approach facilitates more accurate integration with larger time steps, fewer evaluations, and reduces variance in loss gradients during training. Specifically, for a mini-batch $\{(\tau^m, c^m)\}_{m=1}^M$, we construct $x_1^m$ deterministically from $(s_{start}^m, s_{goal}^m)$, then solve a Sinkhorn-Knopp problem Cuturi (2013) over the cost matrix $C_{ij} = \|\tau^i - x_1^j\|_2^2$ to obtain an optimal coupling $\boldsymbol{P}^{OT}$. Sampling pairs $\{(\tau^{i^m}, x_1^{j^m})\}_{m=1}^M$ from this coupling effectively straightens the learned flow, reducing the effective Lipschitz constant of the drift. As rigorously proven by Theorem E.2 and Corollary E.3, this OT coupling provably tightens the FM velocity Lipschitz constant and improves the coarse-step stability condition (Theorem E.1), allowing for larger coarse steps and a smaller coarse budget $K$ for a fixed error target.

**Training Objective.** We optimize a composite objective comprising four complementary loss components, which collectively facilitate the learning of our unified multi-scale planner. This objective couples the straightened conditional path with: (i) a consistency loss $\mathcal{L}_{GCTM}$ enforcing self-consistency across arbitrary time intervals; (ii) a flow-matching anchor $\mathcal{L}_{FM}$ that ties the model to the data distribution; (iii) a multi-scale consistency loss $\mathcal{L}_{MS}$ that enforces compositional consistency across a fixed coarse grid used at inference; and (iv) a disentangled control regularizer $\mathcal{L}_{ctrl}$ that structures the latent space for interpretable control.

Given a sampled expert trajectory $\tau$ from $\mathcal{D}$, and its corresponding start-goal condition $c = (s_{start}, s_{goal})$, we deterministically form $\mathbf{x}_1$ from $c$, as described in the trajectory representation.

**Consistency Objective.** The consistency loss, $\mathcal{L}_{GCTM}$, directly applies the self-consistency principle of GCTMs, ensuring that a one-step transition from time $t$ to $s$ is consistent with a two-step transition via an intermediate time $u$. This loss, derived from Eq. 9, is expressed as:

$$\mathcal{L}_{GCTM}(\theta) = \mathbb{E}_{t,s,u,(\tau,\mathbf{x}_1,c)\sim\mathcal{D}} \left[ d\left( G_\theta(\mathbf{x}_t,t,s;c), G_{sg(\theta)}(\mathbf{x}_{t\to u},u,s;c) \right) \right] \tag{11}$$

where $\mathbf{x}_t = (1-t)\tau + t\mathbf{x}_1$ is the linearly interpolated sample at time $t$.

**Flow Matching Objective.** Complementary to the consistency objective, the flow-matching loss, $\mathcal{L}_{FM}$, serves as an anchor, tying the model to the ground truth data distribution by training $g_\theta$ as an effective denoiser. It is expressed as:

$$\mathcal{L}_{FM}(\theta) = \mathbb{E}_{t,\tau}\mathbb{E}_{\mathbf{x}_t|\tau} \left[ \|\tau - g_\theta(\mathbf{x}_t,t,t)\|_2^2 \right] \tag{12}$$

where $g_\theta$ is the neural network component of the GCTM parameterization $G_\theta(\mathbf{x}_t,t,s) = \frac{s}{t}\mathbf{x}_t + (1-\frac{s}{t})g_\theta(\mathbf{x}_t,t,s)$.

**Multi-Scale Consistency Objective.** A key insight of our work is that a single model can be queried at different temporal scales to produce a planning hierarchy. To explicitly enforce this crucial property during training, we introduce a *multi-scale consistency loss*. Let $\mathcal{T}_c = \{t_0, t_1, \ldots, t_K\}$ be a fixed, discrete set of coarse time steps used for high-level planning at inference, with $1 = t_0 > t_1 > \cdots > t_K > 0$. This loss term penalizes inconsistencies between a long temporal jump and a sequence of shorter jumps across these specific scales (all conditioned on $c$):

$$\mathcal{L}_{MS}(\theta) = \sum_{i=0}^{K-1} \mathbb{E}_{\mathbf{x}_{t_0}} \left[ d\left( G_\theta(\mathbf{x}_{t_0},t_0,t_{i+1};c), G_\theta(G_\theta(\mathbf{x}_{t_0},t_0,t_i;c),t_i,t_{i+1};c) \right) \right] \tag{13}$$

This objective directly encourages the model to produce coherent plans across the hierarchy, and in particular reduces the semigroup-residual term that appears as $K \cdot R_\theta$ in the bound of Theorem E.1.

**Disentangled Control Objective.** To enhance the controllability of our planner, we introduce a regularization term that encourages disentangled control over the plan's characteristics via the latent code $\epsilon$. As detailed in Section 4.2 and Appendix F, our goal is to ensure that different dimensions of the latent code affect orthogonal aspects of the final trajectory $\tau$. We achieve this by penalizing the dot product of the gradients of the trajectory with respect to different latent dimensions:

$$\mathcal{L}_{ctrl}(\theta) = \mathbb{E}_\epsilon \sum_{i\neq j} \left\| \left(\frac{\partial\tau}{\partial\epsilon_i}\right)^T \left(\frac{\partial\tau}{\partial\epsilon_j}\right) \right\|^2 \tag{14}$$

The final training objective is a weighted sum of all four losses:

$$\mathcal{L}(\theta) = \mathcal{L}_{GCTM}(\theta) + \lambda_{FM}\mathcal{L}_{FM}(\theta) + \lambda_{MS}\mathcal{L}_{MS}(\theta) + \lambda_{ctrl}\mathcal{L}_{ctrl}(\theta) \tag{15}$$

This combined objective ensures that the model learns a meaningful, hierarchically consistent, and controllable representation of the entire trajectory space, enabling robust and adaptable long-horizon planning. The pseudocode for the complete training algorithm, Alg. 1, is provided in Appendix B.

## 4.2 INFERENCE PROCESS

Our inference process is a single-model, multi-scale procedure where a planning hierarchy seamlessly emerges by querying the same GCTM at coarse and fine temporal resolutions. The full procedure is detailed in Alg. 2.

**Coarse planning via vectorized multi-time queries.** Given a start–goal query $c = (s_{start}, s_{goal})$, we initialize at $t = 1$ with a condition-dependent embedding $\mathbf{x}_1$. We then efficiently compute all coarse states $\{t_k\}_{k=1}^{K}$ in a single batched forward pass, leveraging a shared model trunk and lightweight heads. The stability of these vectorized multi-time queries is ensured as the composition error between direct and chained evaluations is controlled by the semigroup-residual term $R_\theta$ in Theorem E.1:

$$\mathbf{x}_{t_k} = G_\theta(\mathbf{x}_1,1,t_k;c), \qquad 1 > t_1 > \cdots > t_K > 0. \tag{16}$$

Optionally, each coarse state can be decoded to a clean trajectory $\hat{\tau}_k = g_\theta(\mathbf{x}_{t_k},t_k,t_k;c)$ to extract human-interpretable subgoals for visualization.

**Fine-grained synthesis within intervals.** For each high-level interval $(t_{k+1}, t_k]$, fine-grained motion is synthesized by vectorizing numerous fine-time evaluations using the same GCTM. This process efficiently generates smooth trajectory segments with reduced computational cost:

$$\mathbf{x}_{k,j} = G_\theta(\mathbf{x}_{t_k}, t_k, t'_j; c), \quad t_k > t'_j > t_{k+1}. \tag{17}$$

Decoding these fine-grained evaluations yields dense trajectory segments $\tau_k$, which are then concatenated to form the complete final trajectory $\tau_{final} = \tau_0 \circ \tau_1 \circ \cdots \circ \tau_K$. Our multi-scale consistency objective, enforced during training, explicitly promotes compositional coherence between these synthesized segments and their subsequent subgoals, ensuring a fluid and consistent overall plan.

**Candidate generation, scoring, and refinement.** To enhance the diversity and quality of the generated plans, we employ a candidate generation and refinement process. We first sample $N$ distinct latent codes, which are used to perturb the initial condition-dependent embedding $\mathbf{x}_1$, thereby inducing diverse candidate trajectories. Each candidate is then generated by repeating the coarse planning and fine-grained synthesis steps described above. These candidates are subsequently scored based on a terminal error metric (e.g., distance to the goal state). To further improve the best candidates, we refine the top-$m$ performing trajectories by applying a few steps of gradient descent in their respective latent spaces, optimizing directly on the terminal objective. Finally, the best-scoring trajectory from the refined set is returned as the planner's output.

## 5 EXPERIMENTS

In this section, we present empirical results that validate the effectiveness of our proposed MCPlanner algorithm. We evaluate our method on a range of challenging offline goal-conditioned tasks.

### 5.1 EXPERIMENTAL SETUP

**Environments.** We evaluate our proposed algorithm on the recently proposed OGBench Park et al. (2025), a benchmark designed to evaluate algorithms in offline goal-conditioned RL across different tasks and datasets. We employ environments from 4 locomotion (PointMaze, AntMaze, Humanoid-Maze, AntSoccer), and 3 manipulation (Cube, Puzzle, Scene) domains. More details about these tasks and offline datasets are provided in the Appendix A and Implementation details are deferred to Appendix B

**Baselines.** We compare MCPlanner with prior state-of-the-art diffusion planners like Diffuser (Janner et al., 2022), Decision Diffuser (DD) (Ajay et al., 2023), AdaptDiffuser (Liang et al., 2023), DiffuserLite (Dong et al., 2024), Diffusion Veteran (Lu et al., 2025); and hierarchical diffusion planners like HDMI (Li et al., 2023) and SHD (Chen et al., 2024).

### 5.2 RESULTS

**Q: How does MCPlanner compare to the baselines on Offline Goal-conditioned RL tasks?**

**A:** Table 1 presents a comprehensive comparison of MCPlanner against state-of-the-art diffusion-based planners on the OGBench benchmark. The results demonstrate a clear and consistent advantage for our method. MCPlanner achieves the highest success rates across all tasks and datasets, often by a substantial margin.

**Dominance in Locomotion:** In the challenging locomotion tasks (`pointmaze`, `antmaze`, `humanoidmaze`, and `antsoccer`), MCPlanner consistently outperforms all baselines. Notably, the performance gap widens on more complex, long-horizon environments such as the 'large' and 'giant' mazes. This highlights the effectiveness of our unified hierarchical approach for long-horizon planning.

**Superior Performance in Manipulation:** The strong performance extends to manipulation tasks (`cube`, `scene`, `puzzle`). Across these diverse environments, MCPlanner demonstrates superior planning capabilities. This consistent state-of-the-art performance across nearly all tasks underscores the robustness and generalizability of the MCPlanner framework.

### 5.3 ABLATION STUDIES

**Q: Why not employ a traditional two-model hierarchy, as commonly adopted by hierarchical diffusion planners?**

Table 1: Experimental results for the tasks we considered across diverse datasets. The table reports the average binary success rate (%) across five test-time goals for each task, averaged over 8 seeds. Standard deviations are indicated by the $\pm$ symbol. Entries within 95% of the best-performing value in each row are highlighted in **bold**.

| Environment | Dataset Type | Dataset | Diffuser | DD | HDMI | AD | SHD | DL | DV | Ours |
|---|---|---|---|---|---|---|---|---|---|---|
| pointmaze | navigate | pointmaze-medium-navigate-v0 | 29 ±7 | 37 ±4 | 51 ±8 | 65 ±6 | 66 ±5 | 72 ±7 | 79 ±5 | **86** ±2 |
| | | pointmaze-large-navigate-v0 | 18 ±3 | 21 ±5 | 29 ±6 | 37 ±7 | 35 ±6 | 58 ±5 | 74 ±7 | **81** ±5 |
| | | pointmaze-giant-navigate-v0 | 7 ±4 | 11 ±2 | 27 ±4 | 18 ±5 | 31 ±3 | 46 ±2 | 52 ±1 | **68** ±2 |
| | | pointmaze-teleport-navigate-v0 | 15 ±1 | 20 ±5 | 23 ±6 | 25 ±7 | 28 ±1 | 32 ±4 | **43** ±4 | **45** ±7 |
| | stitch | pointmaze-medium-stitch-v0 | 23 ±7 | 35 ±8 | 46 ±2 | 59 ±5 | 64 ±9 | 69 ±6 | 71 ±8 | **81** ±7 |
| | | pointmaze-large-stitch-v0 | 15 ±4 | 25 ±6 | 36 ±5 | 42 ±8 | 48 ±7 | 55 ±4 | 62 ±6 | **73** ±5 |
| | | pointmaze-giant-stitch-v0 | 5 ±2 | 9 ±3 | 19 ±5 | 23 ±4 | 29 ±6 | 38 ±5 | 45 ±3 | **59** ±4 |
| | | pointmaze-teleport-stitch-v0 | 12 ±3 | 18 ±5 | 21 ±4 | 23 ±6 | 26 ±5 | 30 ±3 | **39** ±4 | **42** ±6 |
| antmaze | navigate | antmaze-medium-navigate-v0 | 15 ±5 | 22 ±4 | 31 ±6 | 40 ±5 | 42 ±7 | 51 ±6 | 60 ±4 | **72** ±3 |
| | | antmaze-large-navigate-v0 | 8 ±3 | 14 ±4 | 22 ±5 | 28 ±6 | 31 ±5 | 40 ±4 | 51 ±5 | **65** ±1 |
| | | antmaze-giant-navigate-v0 | 2 ±1 | 5 ±2 | 11 ±4 | 15 ±3 | 19 ±4 | 27 ±5 | 35 ±4 | **48** ±3 |
| | | antmaze-teleport-navigate-v0 | 6 ±2 | 10 ±3 | 15 ±4 | 18 ±5 | 22 ±4 | 26 ±3 | 33 ±5 | **41** ±4 |
| | stitch | antmaze-medium-stitch-v0 | 12 ±4 | 19 ±5 | 28 ±6 | 35 ±4 | 38 ±6 | 47 ±5 | 55 ±3 | **68** ±4 |
| | | antmaze-large-stitch-v0 | 6 ±3 | 11 ±4 | 19 ±5 | 25 ±3 | 28 ±5 | 36 ±4 | 47 ±6 | **61** ±3 |
| | | antmaze-giant-stitch-v0 | 1 ±1 | 4 ±2 | 9 ±3 | 13 ±4 | 17 ±3 | 24 ±4 | 31 ±5 | **44** ±4 |
| | | antmaze-teleport-stitch-v0 | 4 ±2 | 8 ±3 | 13 ±4 | 16 ±3 | 20 ±4 | 24 ±3 | 30 ±4 | **38** ±5 |
| humanoidmaze | navigate | humanoidmaze-medium-navigate-v0 | 5 ±2 | 9 ±3 | 15 ±4 | 21 ±5 | 24 ±4 | 30 ±3 | 38 ±5 | **51** ±4 |
| | | humanoidmaze-large-navigate-v0 | 2 ±1 | 5 ±2 | 9 ±3 | 13 ±4 | 16 ±3 | 22 ±4 | 29 ±3 | **42** ±5 |
| | | humanoidmaze-giant-navigate-v0 | 0 ±0 | 1 ±1 | 3 ±2 | 5 ±2 | 7 ±3 | 11 ±4 | 16 ±3 | **25** ±4 |
| | stitch | humanoidmaze-medium-stitch-v0 | 4 ±2 | 7 ±3 | 13 ±4 | 19 ±3 | 22 ±5 | 28 ±4 | 35 ±3 | **48** ±4 |
| | | humanoidmaze-large-stitch-v0 | 1 ±1 | 3 ±2 | 7 ±3 | 11 ±4 | 14 ±3 | 20 ±3 | 26 ±4 | **39** ±5 |
| | | humanoidmaze-giant-stitch-v0 | 0 ±0 | 1 ±1 | 2 ±1 | 4 ±2 | 6 ±3 | 12 ±7 | 14 ±3 | **16** ±4 |
| antsoccer | navigate | antsoccer-arena-navigate-v0 | 10 ±4 | 15 ±5 | 25 ±6 | 35 ±4 | 38 ±6 | 45 ±5 | 55 ±3 | **68** ±4 |
| | | antsoccer-medium-navigate-v0 | 8 ±3 | 12 ±4 | 20 ±5 | 28 ±3 | 31 ±5 | 38 ±4 | 48 ±6 | **61** ±4 |
| | stitch | antsoccer-arena-stitch-v0 | 9 ±3 | 14 ±4 | 23 ±5 | 32 ±4 | 35 ±6 | 42 ±5 | 52 ±3 | **65** ±4 |
| | | antsoccer-medium-stitch-v0 | 7 ±2 | 11 ±3 | 18 ±4 | 25 ±3 | 28 ±5 | 35 ±4 | 45 ±6 | **58** ±3 |
| cube | play | cube-single-play-v0 | 5 ±2 | 8 ±3 | 12 ±4 | 18 ±5 | 20 ±4 | 25 ±3 | 32 ±5 | **45** ±4 |
| | | cube-double-play-v0 | 2 ±1 | 4 ±2 | 7 ±3 | 11 ±4 | 13 ±3 | 17 ±4 | 23 ±3 | **35** ±5 |
| | | cube-triple-play-v0 | 1 ±1 | 2 ±1 | 4 ±2 | 6 ±3 | 8 ±2 | 11 ±3 | 14 ±4 | **17** ±3 |
| | | cube-quadruple-play-v0 | **0** ±0 | **0** ±0 | **0** ±0 | **0** ±0 | **0** ±0 | **0** ±0 | **0** ±0 | **0** ±0 |
| scene | play | scene-play-v0 | 10 ±4 | 16 ±4 | 24 ±5 | 33 ±4 | 37 ±6 | 45 ±5 | 56 ±3 | **70** ±4 |
| puzzle | play | puzzle-3x3-play-v0 | 8 ±3 | 13 ±4 | 20 ±5 | 28 ±3 | 31 ±5 | 39 ±4 | 49 ±6 | **62** ±5 |
| | | puzzle-4x4-play-v0 | 0 ±0 | 0 ±0 | 0 ±0 | 2 ±3 | 5 ±4 | 11 ±3 | 19 ±5 | **32** ±4 |
| | | puzzle-4x5-play-v0 | 0 ±0 | 0 ±0 | 0 ±0 | 0 ±0 | 5 ±3 | 6 ±3 | 8 ±4 | **16** ±5 |
| | | puzzle-4x6-play-v0 | 0 ±0 | 0 ±0 | 0 ±0 | 0 ±0 | 0 ±0 | 0 ±0 | 6 ±4 | **14** ±3 |

**A:** To investigate this, we compare our unified MCPlanner against a variant, **MCPlanner-2**, which mimics the traditional hierarchical setup. This variant consists of two separately trained GCTMs: a high-level planner that generates a sequence of sparse subgoals, and a low-level planner that synthesizes dense trajectories to connect them. Our experiments, summarized in Table 2, reveal a significant performance drop with the two-model approach. We attribute this gap to two primary factors: (i) coherence gap, and (ii) compounding errors. Therefore, the unified architecture of MCPlanner is not merely a simplification but a crucial design choice that leads to more coherent, robust, and effective long-horizon planning.

**Q: Is the explicit Multi-Scale Consistency Objective $\mathcal{L}_{MS}$ truly essential for performance, or can the GCTM naturally learn this coherence?**

**A:** While the standard GCTM objective (Eq. 11) encourages consistency over arbitrary time intervals, we hypothesized that explicitly enforcing this property on the fixed, coarse time grid used at inference would be beneficial. To validate this, we trained a variant, **MCPlanner w/o $\mathcal{L}_{MS}$**, which omits the multi-scale consistency loss term (Eq. 13). The results, presented in Table 2, confirm our hypothesis. While the model without $\mathcal{L}_{MS}$ still performs reasonably well, there is a consistent performance degradation across all tasks. This suggests that while the base GCTM objective provides a degree of implicit consistency, it is not sufficient to guarantee the strong compositional coherence required for our hierarchical inference scheme.

**Q: Does the Conditional Optimal Transport (OT) coupling offer a significant advantage over simpler, independent couplings for flow straightening?**

**A:** As established in our theoretical framework (Theorem E.2), using an OT coupling is designed to straighten the learned flow between the initial and final trajectory distributions. This leads to a smaller Lipschitz constant for the ODE drift, which in turn allows for more stable training and more accurate integration with fewer steps. To quantify this benefit, we trained a variant, **MCPlanner w/o OT**, that uses a simpler random pairing between expert trajectories and their conditional priors within each batch. The results in Table 2 clearly show the practical benefits of OT coupling.

Table 2: Ablation study on the unified model architecture, multi-scale consistency objective, and conditional optimal transport coupling.

| Environment | Dataset | MCPlanner | MCPlanner-2 | w/o $\mathcal{L}_{MS}$ | w/o OT |
|---|---|---|---|---|---|
| pointmaze | medium-navigate-v0 | 86 ±2 | 67 ±4 | 79 ±3 | 81 ±7 |
| | large-navigate-v0 | 81 ±5 | 60 ±9 | 72 ±2 | 75 ±8 |
| | giant-navigate-v0 | 68 ±2 | 45 ±5 | 57 ±5 | 61 ±7 |
| | teleport-navigate-v0 | 45 ±7 | 21 ±2 | 32 ±9 | 39 ±4 |
| antmaze | medium-navigate-v0 | 72 ±3 | 55 ±7 | 66 ±3 | 68 ±8 |
| | large-navigate-v0 | 65 ±1 | 46 ±9 | 57 ±5 | 58 ±2 |
| | giant-navigate-v0 | 48 ±3 | 27 ±1 | 36 ±8 | 42 ±5 |
| | teleport-navigate-v0 | 41 ±4 | 16 ±9 | 31 ±5 | 36 ±3 |
| humanoidmaze | medium-navigate-v0 | 51 ±4 | 36 ±7 | 46 ±2 | 47 ±8 |
| | large-navigate-v0 | 42 ±5 | 25 ±2 | 31 ±3 | 34 ±5 |
| | giant-navigate-v0 | 25 ±4 | 7 ±5 | 12 ±9 | 17 ±1 |

Table 3: Ablation study on the coarse time grid resolution ($K$).

| Environment | Dataset | $K = 2$ | $K = 5$ | $K = 10$ |
|---|---|---|---|---|
| pointmaze | medium-navigate-v0 | 79 ±3 | 86 ±2 | 87 ±2 |
| | large-navigate-v0 | 71 ±2 | 81 ±5 | 81 ±9 |
| | giant-navigate-v0 | 55 ±7 | 68 ±2 | 69 ±5 |
| | teleport-navigate-v0 | 29 ±9 | 45 ±7 | 45 ±6 |
| antmaze | medium-navigate-v0 | 63 ±4 | 72 ±3 | 73 ±8 |
| | large-navigate-v0 | 53 ±2 | 65 ±1 | 66 ±6 |
| | giant-navigate-v0 | 33 ±5 | 48 ±3 | 49 ±2 |
| | teleport-navigate-v0 | 23 ±8 | 41 ±4 | 42 ±5 |
| humanoidmaze | medium-navigate-v0 | 43 ±5 | 51 ±4 | 52 ±1 |
| | large-navigate-v0 | 34 ±8 | 42 ±5 | 42 ±9 |
| | giant-navigate-v0 | 18 ±1 | 25 ±4 | 25 ±6 |

**Q: How does the choice of coarse time grid $\mathcal{T}_c$ affect performance?**

**A:** The coarse time grid $\mathcal{T}_c$ determines the resolution of the high-level plan. A denser grid (larger $K$) allows for more frequent subgoals, potentially improving plan accuracy, but increases computational cost. Conversely, a sparser grid is more efficient but may fail to capture the necessary detail for complex, long-horizon tasks. To analyze this trade-off, we evaluated MCPlanner with three different grid resolutions: a sparse grid with $K = 2$ coarse steps, our default grid with $K = 5$ steps, and a dense grid with $K = 10$ steps. The results, shown in Table 3, highlight the importance of this choice. Our chosen grid with $K = 5$ steps provides enough high-level guidance to solve challenging tasks without incurring the unnecessary computational overhead of an overly dense plan.

**Q: How does the choice of candidate count $N$ affect performance?**

**A:** The number of trajectory candidates, $N$, is a crucial hyperparameter that balances planning performance against computational cost. Generating more candidates allows the planner to explore a wider range of solutions, increasing the probability of finding a successful path, especially in complex environments with multi-modal solutions.

To understand this trade-off, we evaluated MCPlanner with varying numbers of candidates: $N = 1, 4, 16, \& 32$. The results, presented in Figure 3, demonstrate a clear trend. Moving from $N = 1$ to $N = 16$ yields significant performance improvements across all tasks. However, the gains begin to diminish beyond $N = 16$. Increasing the candidate count to $N = 32$ provides only marginal or no improvement, while doubling the generation time. Therefore, our choice of $N = 16$ represents a well-balanced trade-off, maximiz-

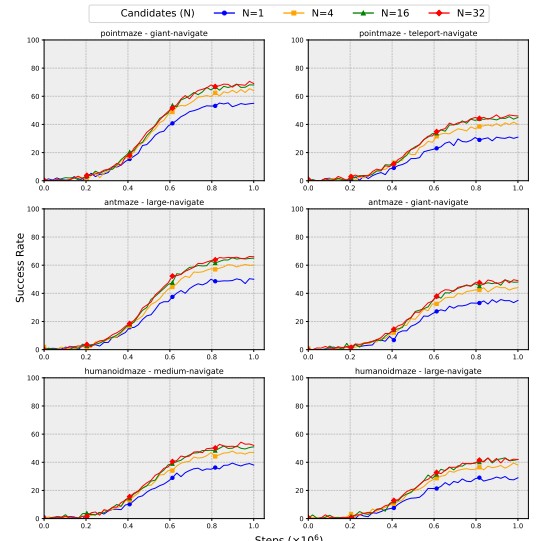

Figure 3: Ablation study on the number of trajectory candidates ($N$).

ing success rates without incurring excessive computational overhead, making the planner both effective and efficient.

## 6 CONCLUSION

In this paper, we introduced MCPlanner, a novel framework that addresses the challenges of long-horizon planning in offline reinforcement learning. We argued that traditional hierarchical methods, which rely on separate models for high-level and low-level planning, often suffer from coherence gaps and increased complexity. MCPlanner overcomes these limitations by employing a single Generalized Consistency Trajectory Model (GCTM) to form a seamless, multi-scale planning hierarchy. By querying the same model at different temporal resolutions, we can generate both coarse, long-horizon subgoals and fine-grained, dense motions within a unified and consistent framework. Extensive experiments on the OGBench benchmark, spanning 35 challenging locomotion and manipulation tasks, demonstrated that MCPlanner consistently outperforms prior state-of-the-art methods.

## REPRODUCIBILITY STATEMENT

To ensure the reproducibility of MCPlanner, we have made significant efforts. For novel models and algorithms, implementation details and hyperparameters are extensively discussed in Appendix B. For theoretical results, clear explanations of all assumptions and complete proofs of claims are provided in Appendix E. For datasets used in the experiments, a complete description of the data processing steps can be found in Appendix A.

## LLM USAGE STATEMENT

We only used Large Language Models (LLMs) to polish the writing and improve the clarity and flow of the text in this paper. No LLMs were used for generating content, ideas, or experimental results.

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

## A   Tasks and Datasets

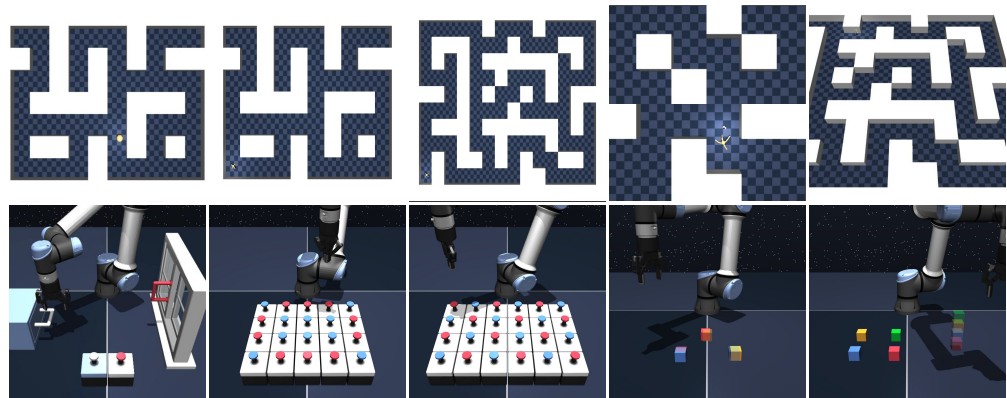

Figure 4: Visualization of a subset of tasks we considered from OGBench.

To validate the effectiveness of MCPlanner across diverse offline goal-conditioned reinforcement learning scenarios, we evaluate our approach on the recently introduced OGBench benchmark Park et al. (2025). This comprehensive benchmark provides a systematic evaluation framework that challenges long-horizon planning algorithms across two distinct domains: locomotion, and manipulation tasks. The benchmark's design particularly emphasizes the planning capabilities that MCPlanner aims to address, including goal stitching, long-horizon reasoning, and the ability to synthesize coherent trajectories from offline data.

### A.1   Locomotion Tasks

**PointMaze (`pointmaze`).** This task involves controlling a 2D point mass agent to navigate through maze environments of varying complexity. We evaluate MCPlanner on four maze configurations: `medium` (the standard baseline), `large` (increased complexity), `giant` (the most challenging layout requiring up to 1500 environment steps), and `teleport` (featuring stochastic teleporters that test robustness to environmental uncertainty). The task provides an ideal testbed for our unified hierarchical planning approach, as successful navigation requires both high-level path planning around obstacles and fine-grained control for precise movement execution.

**AntMaze (`antmaze`).** Building upon PointMaze, this task challenges MCPlanner to control an 8-degree-of-freedom quadrupedal Ant agent through the same maze layouts. The increased morphological complexity introduces additional planning challenges, as the agent must coordinate multiple joints while maintaining global navigation objectives. The longer action sequences required for locomotion make this task particularly suitable for evaluating our multi-scale consistency approach, which ensures coherence between coarse waypoints and fine-grained motion synthesis.

**HumanoidMaze (`humanoidmaze`).** The most complex locomotion task involves controlling a 21-DoF Humanoid agent through maze environments. This task represents the ultimate test of our planning framework's ability to handle high-dimensional action spaces and complex dynamics. The humanoid's sophisticated morphology requires careful coordination of numerous joints, making long-horizon planning extremely challenging. In the most difficult `giant` configuration, successful navigation can require up to 3000 environment steps, thoroughly testing MCPlanner's ability to maintain trajectory coherence over extended horizons.

**AntSoccer (`antsoccer`).** This novel locomotion task extends beyond simple navigation by requiring the Ant agent to manipulate a soccer ball while navigating. The task involves dribbling the ball through two environment types: an open `arena` and a structured `medium` maze. This dual objective of navigation and object manipulation tests MCPlanner's ability to coordinate multiple behavioral modes within a single trajectory, making it an excellent benchmark for our unified planning approach.

## A.2 MANIPULATION TASKS

**Cube (`cube`).** These tasks challenge MCPlanner to control a 6-DoF UR5e robot arm for pick-and-place manipulation of cube blocks. We evaluate on four variants with increasing complexity: `single`, `double`, `triple`, and `quadruple`, corresponding to tasks involving 1-4 cubes respectively. The evaluation goals require moving, stacking, swapping, or permuting cube blocks to achieve specified configurations. This task family is particularly valuable for assessing our approach's ability to learn multi-object manipulation behaviors and synthesize long-term plans that may require up to 8 sequential pick-and-place operations.

**Scene (`scene`).** This manipulation task is designed to test sequential reasoning capabilities through interaction with diverse everyday objects including cube blocks, windows, drawers, and button-controlled locks. The task requires MCPlanner to understand object dependencies and execute complex sequential behaviors. For example, certain goals require unlocking a drawer, opening it, placing an object inside, and closing it again. The longest tasks involve up to eight atomic manipulation behaviors, making this an excellent testbed for our hierarchical planning approach that must maintain coherence across extended behavioral sequences.

**Puzzle (`puzzle`).** Perhaps the most challenging manipulation tasks involve solving the "Lights Out" puzzle using the robot arm. These tasks require pressing buttons on 2D grids of varying sizes (`3x3`, `4x4`, `4x5`, and `4x6`) where each button press toggles the colors of the pressed button and its neighbors. The goal is to achieve desired color configurations through strategic button combinations. This task tests MCPlanner's combinatorial reasoning abilities, as the state space grows exponentially with grid size (up to $2^{24}$ states for the largest puzzle). The most complex puzzles require pressing over 20 buttons in precise sequences, thoroughly challenging our approach's long-horizon planning capabilities.

## A.3 DATASET CHARACTERISTICS

The OGBench benchmark provides multiple dataset types that pose distinct challenges for offline planning algorithms:

**Navigate Datasets.** These datasets consist of trajectories collected by noisy expert policies that navigate environments while reaching randomly sampled goals. They provide diverse coverage of successful behaviors but require MCPlanner to learn from suboptimal demonstrations with varying noise levels.

**Stitch Datasets.** Specifically designed to challenge goal stitching capabilities, these datasets contain only short trajectory segments (e.g., at most 4 cell units in maze tasks). Successful task completion requires MCPlanner to stitch multiple trajectory segments together, with some tasks requiring up to 8 stitching operations. This directly tests our unified model's ability to generate coherent long-horizon plans from fragmented demonstration data.

**Play Datasets.** Used for manipulation tasks, these datasets are collected by non-Markovian expert policies with temporally correlated noise, mimicking realistic data collection scenarios. The unstructured nature of these datasets challenges MCPlanner to extract meaningful behavioral patterns and synthesize novel goal-directed trajectories from seemingly random interactions.

The diversity of these datasets, spanning 35 total configurations across all task domains, provides a comprehensive evaluation framework for assessing MCPlanner's robustness and generalization capabilities across varied data quality and structure conditions.

## B  IMPLEMENTATION DETAILS

This section provides comprehensive details regarding the implementation of MCPlanner, including the specific hyperparameters used in all experiments, the full training algorithm, and the complete inference procedure.

### B.1  HYPERPARAMETERS

Table 4: MCPlanner hyperparameters used in all experiments.

| Parameter | Setting |
|---|---|
| Backbone | 1D U-Net (based on (Song et al., 2021)'s architecture) |
| Horizon $H$ | 32 |
| Model width | 64 |
| Positional timestep embeddings | Yes |
| Kernel size | 5 |
| $\lambda_{\text{FM}}$ | 0.1 |
| $\lambda_{\text{MS}}$ | 1.0 |
| $\lambda_{\text{ctrl}}$ | 0.1 |
| Coarse time grid $\mathcal{T}_c$ | $\{1.0, 0.8, 0.6, 0.4, 0.2, 0.0\}$ |
| EMA rate | 0.999 |
| EDM | $\sigma_{\min} = 0.002$, $\sigma_{\max} = 80.0$ |
| Number of discretization steps $N$ | Doubles every 100k iterations, starting from 4 |
| Time $\hat{t}$ distribution | Beta(3,1) |
| Distance $d$ (for $\mathcal{L}_{GCTM}$) | Pseudo-huber loss with $c = 0.00054\sqrt{d}$ |
| Gaussian perturbation for $\mathbf{x}_1$ | $\mathcal{N}(0, 0.05^2\mathbf{I})$ |
| Integrator | Second order Heun |
| Batch size | 64 |
| Training steps | $10^6$ |
| Candidates $N$ | 16 |
| Latent refinement steps $r$ | 2 |
| Top-$m$ candidates | 4 |
| Reranking | Lightweight critic |

In this section, we provide detailed explanations for the hyperparameters and design choices listed in Table 4, aligning with best practices for reproducibility and clarity, similar to the GCTM framework.

**Bootstrapping Scores.** In all our experiments, we train GCTMs without a pre-trained score model. Analogous to CTMs, we utilize velocity estimates provided by an exponential moving average (EMA) of the model parameters $\theta_{EMA}$ to solve ODEs, with an EMA decay rate of 0.999.

**Time Discretization.** We discretize the unit interval into a finite number of timesteps $\{t_n\}_{n=0}^N$ where $t_0 = 0 < t_1 < \cdots < t_N = 1$. This is based on the EDM (Karras et al., 2022) schedule, which solves the PFODE on $(\sigma_{\min}, \sigma_{\max})$ for $0 < \sigma_{\min} < \sigma_{\max}$ with $\rho = 7$. We convert this to the FM ODE discretization using $t_n = \sigma_n/(1 + \sigma_n)$, where we fix $\sigma_{\min} = 0.002$. We note that $\sigma_{\max}$ controls the emphasis on time near $t = 1$, with larger $\sigma_{\max}$ placing more discretization points closer to $t = 1$.

**Number of Discretization Steps $N$.** Unlike CTMs, which use a fixed $N = 18$, our approach doubles $N$ every $100k$ iterations, starting from $N = 4$.

**Time $\hat{t}$ Distribution.** For the training of our GCTM-based planner, we sample $\hat{t} \sim \text{beta}(3, 1)$. This distribution places higher emphasis on sampling intermediate time points relevant for learning trajectory segments.

**Network Conditioning.** We employ the EDM conditioning scheme, which has shown robust performance in diffusion-based generative models, following the practices established in CTMs.

**Distance $d$ (for $\mathcal{L}_{GCTM}$).** For the consistency loss $\mathcal{L}_{GCTM}$, we utilize the pseudo-huber loss, defined as $d(\mathbf{x}_t, \hat{\mathbf{x}}_t) = \sqrt{\|\mathbf{x}_t - \hat{\mathbf{x}}_t\|_2^2 + c^2} - c$, where $c = 0.00054\sqrt{d}$ and $d$ is the dimension of $\mathbf{x}_t$. This choice provides a robust measure of similarity between trajectory samples.

**Gaussian Perturbation for $\mathbf{x}_1$.** To ensure diversity and enable one-to-many generation, we apply a Gaussian perturbation to the $\mathbf{x}_1$ samples, drawing from a normal distribution multiplied by $0.05$. This acts as a latent source of randomness, allowing the GCTM network to map conditions to distinct trajectories.

**ODE Solver.** We use the second-order Heun solver for numerically integrating the ODEs and calculating terms within the $\mathcal{L}_{GCTM}(\theta)$ objective.

**Batch Size, Optimizer, $\lambda_{\text{FM}}$, and Network Architecture.** We use a batch size of 64 for all experiments. The Adam optimizer (Kingma & Ba, 2017) is employed with a learning rate of $\eta = 0.0002/(128/\texttt{batch\_size})$ and default $(\beta_1, \beta_2) = (0.9, 0.999)$. The coefficient for the Flow Matching loss is set to $\lambda_{\text{FM}} = 0.1$. Our network is a modified **SongUNet** (Song et al., 2021) that accepts two time conditions $t$ and $s$ via two time embedding layers.

## B.2 ALGORITHMS

---

**Algorithm 1** MCPlanner Training

---

1: **Require:** Dataset $\mathcal{D}$; GCTM $G_\theta$, denoiser head $g_\theta$; coefficients $\lambda_{FM}, \lambda_{MS}, \lambda_{ctrl}$; coarse steps $\mathcal{T}_c$; OT regularization $\tau$; EMA rate $\alpha$; integrator order $p$.
2: **while** not converged **do**
3:      $\{(\tau_i, s_{start,i}, s_{goal,i})\}_{i=1}^B \sim \mathcal{D}$
4:      $x_i^{prior} \leftarrow \text{LinearInterp}(s_{start,i} \rightarrow s_{goal,i})$ with zero actions
5:      **function** CONDITIONAL OT PAIRING
6:          $C_{ij} \leftarrow \|\tau_i - x_j^{prior}\|_2^2$
7:          $\text{Sinkhorn}(C, \tau)$ to obtain $\boldsymbol{P}^{OT}$
8:          $\{(i_m, j_m)\}_{m=1}^B \sim \boldsymbol{P}^{OT}$
9:          $\mathbf{x}_{0,m} \leftarrow \tau_{i_m}, \mathbf{x}_{1,m} \leftarrow x_{j_m}^{prior}, c_m \leftarrow (s_{start,j_m}, s_{goal,j_m})$
10:         Sample times $t, s, u, \hat{t} \sim \text{Unif}(0,1)$ (independently per $m$)
11:         $\mathbf{x}_{t,m} \leftarrow (1-t)\mathbf{x}_{0,m} + t\mathbf{x}_{1,m}$
12:         $\mathbf{x}_{\hat{t},m} \leftarrow (1-\hat{t})\mathbf{x}_{0,m} + \hat{t}\mathbf{x}_{1,m}$
13:      **end function**
14:      $\mathcal{L}_{FM} \leftarrow \frac{1}{B}\sum_m \|\mathbf{x}_{0,m} - g_\theta(\mathbf{x}_{\hat{t},m}, \hat{t}, \hat{t}; c_m)\|_2^2$
15:      $\widetilde{\mathbf{x}}_{s,m} \leftarrow G_{\text{EMA}}(\mathbf{x}_{t\rightarrow u,m}, u, s; c_m)$
16:      $\mathcal{L}_{GCTM} \leftarrow \frac{1}{B}\sum_m d\big(G_\theta(\mathbf{x}_{t,m}, t, s; c_m), \widetilde{\mathbf{x}}_{s,m}\big)$
17:      $\mathcal{L}_{\text{multi-scale}} \leftarrow \sum_i d\big(G_\theta(\mathbf{x}_{t_0}, t_0, t_{i+1}; c_m), G_\theta(G_\theta(\mathbf{x}_{t_0}, t_0, t_i; c_m), t_i, t_{i+1}; c_m)\big)$
18:      Estimate $\mathcal{L}_{\text{control}}$ via Hutchinson-style Jacobian orthogonality
19:      Total loss: $\mathcal{L} \leftarrow \mathcal{L}_{GCTM} + \lambda_{FM}\mathcal{L}_{FM} + \lambda_{MS}\mathcal{L}_{\text{multi-scale}} + \lambda_{ctrl}\mathcal{L}_{\text{control}}$
20:      $\theta \leftarrow \theta - \eta\nabla_\theta\mathcal{L}$
21:      $\theta_{\text{EMA}} \leftarrow \alpha\theta_{\text{EMA}} + (1-\alpha)\theta$
22: **end while**

---

---

**Algorithm 2** MCPlanner Inference

---

1: **Require:** Trained $G_\theta, g_\theta$
2: **Input:** $s_{start}, s_{goal}$; $c = (s_{start}, s_{goal})$
3: **Params:** Coarse steps $\mathcal{T}_c = \{t_0, \ldots, t_K\}$; fine schedules $\{\mathcal{T}_k^{\text{fine}}\}_{k=0}^{K-1}$; candidate count $N$; refinement steps $r$; step size $\eta$; guidance weight $\beta$
4: $\mathbf{x}_1 \leftarrow \text{LinearInterp}(s_{start} \to s_{goal})$ with zero actions
5: Initialize candidate set $\mathcal{C} \leftarrow \emptyset$
6: **for** $n = 1$ to $N$ **do** $\quad\quad\quad\quad\quad\quad\quad\quad\quad\quad\quad\quad\quad\quad\quad\quad\quad\quad$ ▷ Candidate generation
7: $\quad \epsilon_n \sim \mathcal{N}(0, \mathbf{I})$
8: $\quad \mathbf{x}_1^{(n)} \leftarrow \mathbf{x}_1 + \gamma \, \epsilon_n$
9: $\quad \{\mathbf{x}_{t_k}^{(n)}\}_{k=1}^K \leftarrow G_\theta(\mathbf{x}_1^{(n)}, 1, \{t_k\}; c)$
10: $\quad \hat{\tau}_k^{(n)} \leftarrow g_\theta(\mathbf{x}_{t_k}^{(n)}, t_k, t_k; c)$; select $z_k^{(n)}$ from $\hat{\tau}_k^{(n)}$ $\quad\quad\quad\quad$ ▷ optional subgoal extraction
11: $\quad \tau_{final}^{(n)} \leftarrow [\,]$
12: $\quad \mathbf{x}_{t_0}^{(n)} \leftarrow \mathbf{x}_1^{(n)}$
13: $\quad$ **for** $k = 0$ to $K - 1$ **do** $\quad\quad\quad\quad\quad\quad\quad\quad\quad\quad\quad\quad$ ▷ Fine synthesis per interval
14: $\quad\quad \{\mathbf{x}_{k,j}^{(n)}\}_{t' \in \mathcal{T}_k^{\text{fine}}} \leftarrow G_\theta(\mathbf{x}_{t_k}^{(n)}, t_k, \mathcal{T}_k^{\text{fine}}; c)$
15: $\quad\quad$ Decode to dense segment $\tau_k^{(n)}$ and append to $\tau_{final}^{(n)}$
16: $\quad$ **end for**
17: $\quad$ Score: $e_n \leftarrow \|s_T(\tau_{final}^{(n)}) - s_{goal}\|^2$
18: $\quad$ Total score $s_n \leftarrow e_n - \beta v_n$; add $(\epsilon_n, \tau_{final}^{(n)}, s_n)$ to $\mathcal{C}$
19: **end for**
20: Select top-$m$ elements of $\mathcal{C}$ by $s_n$ (e.g., $m = 4$)
21: **for** each selected candidate $(\epsilon, \tau, s)$ **do**
22: $\quad$ **for** $t = 1$ to $r$ **do** $\quad\quad\quad\quad\quad\quad\quad\quad\quad\quad\quad\quad$ ▷ Latent refinement (few steps)
23: $\quad\quad \epsilon \leftarrow \epsilon - \eta \, \nabla_\epsilon \|s_T(\tau_{final}(\epsilon)) - s_{goal}\|^2$ $\quad$ (backprop through $G_\theta$)
24: $\quad$ **end for**
25: $\quad$ Recompute $\tau$ and score $s$; update candidate in $\mathcal{C}$
26: **end for**
27: **return** trajectory with the best (lowest) final score in $\mathcal{C}$

---

# C BASELINES

In order to better validate the performance of our method, we re-implement the following baselines on the OGBench benchmark.

**Diffuser.** (Janner et al., 2022) Diffuser formulates planning as an iterative denoising process using a diffusion probabilistic model. It trains a trajectory-level diffusion model that predicts all timesteps of a plan simultaneously. The training objective for the $\epsilon$-model is given by:

$$\mathcal{L}_{Diffuser}(\theta) = \mathbb{E}_{t,\epsilon,\mathbf{x}_0}\left[\|\epsilon - \epsilon_\theta(\mathbf{x}_t, t)\|^2\right]$$

where $t$ is the diffusion timestep, $\epsilon$ is the noise target, and $\mathbf{x}_t$ is the trajectory $\mathbf{x}_0$ corrupted with noise $\epsilon$. This framework allows for flexible conditioning through classifier-guided sampling, reinterpreting it as a planning strategy.

**Decision Diffuser.** (Ajay et al., 2023) Decision Diffuser frames offline sequential decision-making as a conditional generative modeling problem, diffusing only over states $\mathbf{x}_t := (s_{t'}, s_{t'+1}, \ldots, s_{t'+H-1})$. Actions are inferred via a separate inverse dynamics model $a_t := f_\phi(s_t, s_{t+1})$. It leverages classifier-free guidance during sampling, with the perturbed noise $\hat{\epsilon}$ defined as:

$$\hat{\epsilon} := \epsilon_\theta(\mathbf{x}_t, \emptyset, t) + \omega(\epsilon_\theta(\mathbf{x}_t, c, t) - \epsilon_\theta(\mathbf{x}_t, \emptyset, t))$$

The combined training objective for the noise model $\epsilon_\theta$ and inverse dynamics model $f_\phi$ is:

$$\mathcal{L}_{DD}(\theta, \phi) := \mathbb{E}_{t,\mathbf{x}_0 \in \mathcal{D}, \beta \sim \text{Bern}(p)}[\|\epsilon - \epsilon_\theta(\mathbf{x}_t, (1-\beta)c + \beta\emptyset, t)\|^2] + \mathbb{E}_{(s,a,s') \in \mathcal{D}}[\|a - f_\phi(s, s')\|^2]$$

where $c$ represents the conditioning variable (e.g., return, constraints, or skills) and $\varnothing$ is a dummy value for unconditional noise.

**AdaptDiffuser.** (Liang et al., 2023) AdaptDiffuser is an evolutionary planning method that enhances diffusion models through a self-evolution process. It iteratively generates diverse synthetic expert data for goal-conditioned tasks, guided by reward gradients. A discriminator then selects high-quality data to fine-tune the diffusion model. The iterative training objective to update the diffusion model $\theta$ at phase $k$ is to minimize the negative log-likelihood of the conditional trajectory distribution, given by:

$$\theta_k^* = \arg\min_\theta -\mathbb{E}_{\hat{\mathbf{x}}_0}[\log p_\theta(\hat{\mathbf{x}}_0|c)]$$

where $\hat{\mathbf{x}}_0$ represents the refined dataset at iteration $k$, and $c$ is the conditioning variable. The process also generates new data $\mathbf{x}_{0,k+1} = \mathcal{G}(\mu_{\theta_k^*}, \Sigma, \nabla_{\mathbf{x}_0}\mathcal{J}(\mu_{\theta_k^*}))$ and refines the dataset $\hat{\mathbf{x}}_{0,k+1} = [\hat{\mathbf{x}}_{0,k}, \mathcal{D}(\widetilde{\mathcal{R}}(\mathbf{x}_{0,k+1}))]$. This self-evolutionary process allows AdaptDiffuser to improve its planning performance and adapt to unseen tasks without requiring additional expert data.

**DiffuserLite.** (Dong et al., 2024) DiffuserLite addresses the slow sampling speed of diffusion planning by introducing a Plan Refinement Process (PRP). Instead of generating full long-horizon trajectories in a single shot, PRP employs a coarse-to-fine-grained hierarchical approach. It plans rough trajectories with key points at intervals and progressively refines the first interval, ignoring redundant distant parts. This process uses $L$ planning levels, where at each level $l$, a diffusion model plans a rough trajectory $\mathbf{x}_t$ with temporal horizon $H_l$ and temporal jump $I_l$. The noise estimator $\epsilon_\theta$ for each level $l$ is optimized by minimizing the following objective, which is similar to the standard diffusion loss but applied to the sub-trajectories:

$$\mathcal{L}_{DL}(\theta_l) = \mathbb{E}_{q_0(\mathbf{x}_0), q(\epsilon), t}[||\epsilon_{\theta_l}(\mathbf{x}_t, t, c) - \epsilon||_2^2]$$

where $\mathbf{x}_t = \alpha_t \mathbf{x}_0 + \sigma_t \epsilon$, and $c$ is the estimated property (e.g., cumulative reward) for the rough trajectory. DiffuserLite also utilizes classifier-free guidance during sampling to achieve conditional generation. The hierarchical refinement reduces computational complexity and significantly increases decision-making frequency.

**Diffusion Veteran.** (Lu et al., 2025) Diffusion Veteran (DV) proposes a simple yet strong diffusion planning baseline for offline reinforcement learning. It identifies key design choices for effective diffusion planning, including the use of Transformer as the denoising network backbone, a separate inverse dynamics model for action generation, and Monte Carlo sampling with selection (MCSS) as the guidance algorithm. DV's training involves three main components:

- A Diffusion Transformer Planner $\epsilon_\theta$ is trained to generate state plans $\mathbf{x}_t$ conditioned on the initial state $s_{start}$, where $M$ is the planning stride. The training objective is to predict the noise.

- A Diffusion Inverse Dynamics model $\epsilon_\omega$ is trained to infer the action $\mathbf{a}_t$ from the current state $s_t$ and the planned next state $s_{t+M}$.

- A Critic $V_\phi$ is trained to predict the accumulated discounted returns $R_t$ for a given state plan.

During inference, DV randomly generates $N$ candidate state plans using $\epsilon_\theta$, selects the best plan based on the critic $V_\phi$, and then uses the inverse dynamics model $\epsilon_\omega$ to extract the action from the current state and the next planned state. This architecture enables robust long-horizon planning.

**HDMI.** (Li et al., 2023) Hierarchical Diffusion for Offline Decision Making (HDMI) proposes a hierarchical trajectory-level diffusion probabilistic model for long-horizon tasks in offline reinforcement learning. It employs a cascade framework with two main components:

- **Reward-Conditional Goal Diffuser**: This component discovers subgoals by conditioning on rewards, facilitating the decomposition of complex tasks into manageable subgoals.

- **Goal-Conditional Trajectory Diffuser**: Given the identified subgoals, this component generates corresponding action sequences to achieve each subgoal.

HDMI utilizes planning-based subgoal extraction and transformer-based diffusion to handle suboptimal data and long-range dependencies. The training objective for both diffusers typically follows

the standard diffusion loss, aiming to predict the noise added to corrupted trajectories, and also incorporates reward or goal conditioning for guided generation. During inference, the Reward-Conditional Goal Diffuser first generates a sequence of subgoals, which are then used by the Goal-Conditional Trajectory Diffuser to synthesize dense action plans between them, effectively tackling long-horizon decision-making tasks.

**SHD.** (Chen et al., 2024) Simple Hierarchical Diffuser (SHD), introduces a two-level hierarchical planning framework built upon diffusion models. It comprises a high-level Sparse Diffuser (SD) for subgoal generation and a low-level Diffuser for fine-grained trajectory synthesis. The high-level SD models subsampled trajectories $\mathbf{x}_0^{SD}$, typically consisting of every $K$-th state-action pair, where subgoals are defined as these sparse states. Both the high-level SD and the low-level Diffuser for segment generation are trained with a standard diffusion noise prediction objective:

$$\mathcal{L}(\theta) = \mathbb{E}_{\mathbf{x}_0,t,\epsilon}\left[\|\epsilon - \epsilon_\theta(\mathbf{x}_t)\|^2\right]$$

This objective trains the model to predict the noise $\epsilon$ added to a corrupted trajectory $\mathbf{x}_t$. Additionally, a separate guidance function $\mathcal{J}_\phi(\mathbf{x}_0)$ is trained for both levels to predict the return $R(\mathbf{x}_0)$ of the respective trajectories (full trajectory for high-level, segment for low-level) using the loss:

$$\mathcal{L}(\phi) = \mathbb{E}_{\mathbf{x}_0,t,\epsilon}\left[\|R(\mathbf{x}_0) - \mathcal{J}_\phi(\mathbf{x}_t)\|^2\right]$$

At inference, the high-level planner generates a sequence of sparse subgoals, which the low-level planner then connects with dense trajectories, using the guidance functions to bias towards high-return paths. This hierarchical approach aims to reduce computational cost and improve generalization for long-horizon tasks.

## D  ADDITIONAL ABLATION STUDIES

In this section, we provide further ablation studies to analyze the impact of key components of our MCPlanner framework: the disentangled control objective and the latent refinement process.

Table 5: Ablation study on the disentangled control objective ($\mathcal{L}_{ctrl}$).

| Environment | Dataset | MCPlanner | w/o $\mathcal{L}_{ctrl}$ |
|---|---|---|---|
| pointmaze | medium-navigate-v0 | $86_{\pm2}$ | $85_{\pm3}$ |
| | large-navigate-v0 | $81_{\pm5}$ | $78_{\pm1}$ |
| | giant-navigate-v0 | $68_{\pm2}$ | $64_{\pm8}$ |
| | teleport-navigate-v0 | $45_{\pm7}$ | $41_{\pm6}$ |
| antmaze | medium-navigate-v0 | $72_{\pm3}$ | $71_{\pm6}$ |
| | large-navigate-v0 | $65_{\pm1}$ | $64_{\pm7}$ |
| | giant-navigate-v0 | $48_{\pm3}$ | $45_{\pm3}$ |
| | teleport-navigate-v0 | $41_{\pm4}$ | $38_{\pm2}$ |
| humanoidmaze | medium-navigate-v0 | $51_{\pm4}$ | $48_{\pm9}$ |
| | large-navigate-v0 | $42_{\pm5}$ | $40_{\pm2}$ |
| | giant-navigate-v0 | $25_{\pm4}$ | $21_{\pm4}$ |

**Q: What is the impact of the disentangled control objective $\mathcal{L}_{ctrl}$ on planning performance?**

**A:** The disentangled control objective (Eq. 14) is designed primarily to structure the latent space for more interpretable control, as detailed in Appendix F. However, we were interested in whether this structural regularization also provides a benefit to the overall planning performance. To test this, we trained a variant, **MCPlanner w/o $\mathcal{L}_{ctrl}$**, that omits this loss term. The results are shown in Table 5. We observe a small but consistent degradation in performance across all tested environments. This suggests that encouraging orthogonality in the latent control dimensions not only improves interpretability but also acts as a useful regularizer, preventing overfitting and leading to slightly more robust and generalizable plans. While not its primary purpose, this secondary benefit further justifies its inclusion in our final model.

**Q: How does latent refinement at inference time contribute to the final plan quality?**

**A:** Our inference procedure includes an optional step to refine the top candidate trajectories by performing gradient descent in their latent space to minimize the terminal state error (Alg. 2). To quantify the benefit of this step, we evaluated MCPlanner's performance with varying numbers of refinement steps: $r = 0$ (no refinement), $r = 2$ (our default), and $r = 4$. The results in Table 6 clearly

Table 6: Ablation study on the number of latent refinement steps ($r$).

| Environment | Dataset | $r = 0$ | $r = 2$ | $r = 4$ |
|---|---|---|---|---|
| pointmaze | medium-navigate-v0 | $81_{\pm 7}$ | $86_{\pm 2}$ | $87_{\pm 5}$ |
| | large-navigate-v0 | $75_{\pm 4}$ | $81_{\pm 5}$ | $82_{\pm 8}$ |
| | giant-navigate-v0 | $61_{\pm 6}$ | $68_{\pm 2}$ | $69_{\pm 5}$ |
| | teleport-navigate-v0 | $39_{\pm 1}$ | $45_{\pm 7}$ | $46_{\pm 3}$ |
| antmaze | medium-navigate-v0 | $66_{\pm 3}$ | $72_{\pm 3}$ | $73_{\pm 9}$ |
| | large-navigate-v0 | $59_{\pm 2}$ | $65_{\pm 1}$ | $66_{\pm 6}$ |
| | giant-navigate-v0 | $41_{\pm 7}$ | $48_{\pm 3}$ | $49_{\pm 2}$ |
| | teleport-navigate-v0 | $35_{\pm 5}$ | $41_{\pm 4}$ | $42_{\pm 1}$ |
| humanoidmaze | medium-navigate-v0 | $45_{\pm 7}$ | $51_{\pm 4}$ | $52_{\pm 6}$ |
| | large-navigate-v0 | $36_{\pm 2}$ | $42_{\pm 5}$ | $43_{\pm 3}$ |
| | giant-navigate-v0 | $19_{\pm 1}$ | $25_{\pm 4}$ | $26_{\pm 4}$ |

demonstrate the value of this process. Disabling refinement ($r = 0$) leads to a significant drop in success rates, highlighting that the initial candidates, while diverse, are not always perfectly aligned with the goal. Applying just two steps of refinement provides a substantial boost in performance across all tasks. Increasing the refinement to five steps yields only marginal further improvements while significantly increasing the computational cost at inference time. Therefore, our choice of $r = 2$ offers a strong balance between plan quality and computational efficiency.

## E    THEORETICAL FRAMEWORK

In this section, we provide a more rigorous theoretical foundation for the MCPlanner framework, particularly focusing on the convergence guarantees of the multi-scale inference process. Our approach is built upon the mathematical underpinnings of Generalized Consistency Trajectory Models (GCTMs) and the analysis of multi-step consistency sampling.

### E.1    GCTM AS A SOLUTION TO THE FLOW MATCHING ODE

Recall that MCPlanner uses a GCTM to learn, for each condition $c = (s_{start}, s_{goal})$, the mapping from a condition embedding distribution $q_c(\mathbf{x}_1) = p_{data}(\phi(c))$ to a full trajectory distribution $q_c(\mathbf{x}_0) = p_{data}(\tau \mid c)$. This is achieved by learning the solution to a probability flow ODE derived from Flow Matching. Given an entropy-regularized optimal transport coupling $q_c(\mathbf{x}_0, \mathbf{x}_1)$ induced by triples $(\tau, s_{start}, s_{goal})$, we define a conditional probability path $q_c(\mathbf{x}_t) = \mathbb{E}_{q_c(\mathbf{x}_0, \mathbf{x}_1)}[\delta_{(1-t)\mathbf{x}_0 + t\mathbf{x}_1}(\mathbf{x}_t)]$. The corresponding ODE is given by:

$$d\mathbf{x}_t = t^{-1}(\mathbf{x}_t - \mathbb{E}_{q_c(\mathbf{x}_0 \mid \mathbf{x}_t)}[\mathbf{x}_0]) \, dt, \quad t \in (0, 1) \tag{18}$$

The GCTM, $G_\theta(\mathbf{x}_t, t, s; c)$, is trained to approximate the solution to this ODE, which transports a sample from time $t$ to time $s$ under condition $c$.

### E.2    MULTI-SCALE INFERENCE AS A MULTI-STEP SAMPLING ALGORITHM

The inference process described in Section 4.2 can be viewed as a specialized multi-step sampling algorithm. The high-level plan generation corresponds to large jumps in time, from $t = 1$ to a sequence of coarse time steps $1 > t_1 > t_2 > \cdots > t_K > 0$. While our method synthesizes the fine-grained trajectory between these steps, the theoretical stability of the overall plan rests on the properties of this coarse, multi-step generation. For analysis, we model each coarse update as a deterministic jump using the learned GCTM,

$$\mathbf{x}_{t_{k-1}} = G_\theta(\mathbf{x}_{t_k}, t_k, t_{k-1}; c), \tag{19}$$

optionally followed by a small additive perturbation $\eta_k \sim \mathcal{N}(0, \sigma_k^2 \mathbf{I})$ to capture numerical or stochastic effects.

### E.3    CONVERGENCE GUARANTEES AND EFFICIENCY OF STRAIGHTENED FLOWS

We now establish a stability result for MCPlanner in the FM/GCTM setting, and show that OT-based flow straightening reduces sample complexity. We introduce the following assumptions.

**Assumption 1** *The FM drift* $b(\mathbf{x}, t) = t^{-1}(\mathbf{x} - \mathbb{E}_{q(\mathbf{x}_0|\mathbf{x}_t=\mathbf{x})}[\mathbf{x}_0])$ *is $L$-Lipschitz in* $\mathbf{x}$ *uniformly in* $t \in (0, 1)$.

**Assumption 2** *The data distribution has finite second moments.* $\mathbb{E}_{p_{data}(\tau)}[\|\tau\|_2^2] \leq m_2 < \infty$.

**Assumption 3** *(i) Denoiser error:* $e_g = \sup_t \mathbb{E}[\|g_\theta(\mathbf{x}_t, t, t; c) - \mathbb{E}[\mathbf{x}_0|\mathbf{x}_t]\|]$. *(ii) Semigroup residual:* $R_\theta = \sup_{t \geq u \geq s} \mathbb{E}[\|G_\theta(\mathbf{x}_t, t, s; c) - G_\theta(G_\theta(\mathbf{x}_t, t, u; c), u, s; c)\|]$. *(iii) Numerical integration error per call* $O(h^p)$.

Under these assumptions, we can state the following theorem regarding the convergence of the generated trajectories.

**Theorem E.1** *Let $\hat{p}$ be the distribution of trajectories generated by MCPlanner after $K$ coarse steps over grid $\{t_k\}$ with maximum step size $h$. Under Assumptions 1, 2, and 3, there exists a constant $C$ depending on $L$ and the moment bound such that*

$$W_2(\hat{p}, p_{data}(\tau)) \leq C\left(K R_\theta + h e_g + h^p + (\textstyle\sum_k \sigma_k^2)^{1/2}\right). \tag{20}$$

*If in addition $p_{data}(\tau)$ satisfies a transport-entropy inequality $T_2(\alpha)$, then*

$$D_{\mathrm{KL}} p_{data}(\tau)\hat{p} \leq \tfrac{1}{2\alpha} W_2^2(\hat{p}, p_{data}(\tau)). \tag{21}$$

*Moreover, when training with an entropy-regularized OT coupling that straightens the conditional path, the effective Lipschitz constant of the drift reduces from $L$ to $\tilde{L} < L$ (in practice measurable via local Jacobian norms), which tightens the bound and permits larger coarse steps $h$ and fewer evaluations $K$ for the same target error.*

**Proof.** Let $\Phi_{t \to s}$ denote the flow map of the FM ODE $\dot{\mathbf{x}}_\tau = b(\mathbf{x}_\tau, \tau)$ from time $t$ to $s$, and write $h_k = t_k - t_{k-1}$. Define the generated update

$$\hat{\mathbf{x}}_{t_{k-1}} = G_\theta(\hat{\mathbf{x}}_{t_k}, t_k, t_{k-1}; c) + \eta_k, \qquad \eta_k \sim \mathcal{N}(0, \sigma_k^2 \mathbf{I}), \tag{22}$$

and the true update $\mathbf{x}_{t_{k-1}}^* = \Phi_{t_k \to t_{k-1}}(\mathbf{x}_{t_k}^*)$. Let $\mu_k$ and $\nu_k$ be the laws of $\hat{\mathbf{x}}_{t_k}$ and $\mathbf{x}_{t_k}^*$, respectively, and define the per-step error

$$E_k := W_2(\mu_k, \nu_k). \tag{23}$$

We first record two standard facts.

(F1) Lipschitz pushforward. If $T$ is $L_T$-Lipschitz, then $W_2(T_{\#}\mu, T_{\#}\nu) \leq L_T W_2(\mu, \nu)$. Under Assumption 1, $\Phi_{t \to s}$ is $\exp(L(t - s))$-Lipschitz in its spatial argument (Gronwall).

(F2) Additive noise. For independent additive noise $\eta$ with $\mathbb{E}\|\eta\|_2^2 = \sigma^2$, $W_2(\mu * \mathcal{L}(\eta), \mu) \leq \sigma$.

Now apply the triangle inequality with $A_k(\cdot) = G_\theta(\cdot, t_k, t_{k-1}; c)$ and $\Phi_k(\cdot) = \Phi_{t_k \to t_{k-1}}(\cdot)$:

$$E_{k-1} = W_2\big((A_k)_{\#}\mu_k * \mathcal{L}(\eta_k), (\Phi_k)_{\#}\nu_k\big) \tag{24}$$

$$\leq W_2\big((A_k)_{\#}\mu_k * \mathcal{L}(\eta_k), (A_k)_{\#}\mu_k\big) + W_2\big((A_k)_{\#}\mu_k, (\Phi_k)_{\#}\mu_k\big) + W_2\big((\Phi_k)_{\#}\mu_k, (\Phi_k)_{\#}\nu_k\big) \tag{25}$$

$$\leq \underbrace{\sigma_k}_{\text{(noise)}} + \underbrace{\delta_k}_{\text{(map error)}} + \underbrace{e^{Lh_k} E_k}_{\text{(pushforward)}}, \tag{26}$$

where

$$\delta_k := \left(\mathbb{E}_{\mathbf{x} \sim \mu_k} \|A_k(\mathbf{x}) - \Phi_k(\mathbf{x})\|_2^2\right)^{1/2}. \tag{27}$$

It remains to bound $\delta_k$ in terms of the approximation errors in Assumption 3. Define the learned drift $\hat{b}(\mathbf{x}, t) = t^{-1}(\mathbf{x} - g_\theta(\mathbf{x}, t, t; c))$ and the associated learned flow $\hat{\Phi}_{t \to s}$ obtained by integrating $\dot{\mathbf{x}}_\tau = \hat{b}(\mathbf{x}_\tau, \tau)$ over $[s, t]$. By variation-of-constants and Gronwall (using Assumption 1),

$$\left\|\hat{\Phi}_{t_k \to t_{k-1}}(\mathbf{x}) - \Phi_{t_k \to t_{k-1}}(\mathbf{x})\right\|_2 \leq e^{Lh_k} \int_{t_{k-1}}^{t_k} \left\|\hat{b}(\mathbf{x}_\tau, \tau) - b(\mathbf{x}_\tau, \tau)\right\|_2 d\tau. \tag{28}$$

Since $\hat{b} - b = t^{-1}\big(\mathbb{E}[\mathbf{x}_0|\mathbf{x}_t] - g_\theta(\mathbf{x}_t, t, t; c)\big)$ and $t \in [t_{k-1}, t_k]$ is bounded away from $0$ on each coarse interval, Assumption 3 implies

$$\mathbb{E}\left\|\hat{\Phi}_{t_k \to t_{k-1}}(\mathbf{x}) - \Phi_{t_k \to t_{k-1}}(\mathbf{x})\right\|_2 \;\le\; C_1 \, h_k \, e_g. \tag{29}$$

Next, $G_\theta(\cdot, t_k, t_{k-1}; c)$ is implemented by a numerical integrator of order $p$ (Heun) and trained to satisfy a semigroup relation. Let $\Psi_{t_k \to t_{k-1}}$ denote one Heun step applied to $\hat{b}$. Then

$$\mathbb{E}\left\|G_\theta(\mathbf{x}, t_k, t_{k-1}; c) - \Psi_{t_k \to t_{k-1}}(\mathbf{x})\right\|_2 \;\le\; C_2 \, R_\theta, \qquad \mathbb{E}\left\|\Psi_{t_k \to t_{k-1}}(\mathbf{x}) - \hat{\Phi}_{t_k \to t_{k-1}}(\mathbf{x})\right\|_2 \;\le\; C_3 \, h_k^p. \tag{30}$$

Combining equation 29 and equation 30 with triangle inequality yields

$$\delta_k \;=\; \left(\mathbb{E}\left\|G_\theta(\mathbf{x}, t_k, t_{k-1}; c) - \Phi_{t_k \to t_{k-1}}(\mathbf{x})\right\|_2^2\right)^{1/2} \;\le\; C\left(R_\theta + h_k e_g + h_k^p\right). \tag{31}$$

Substituting equation 31 into the recursion equation 26 and unrolling over $k = K, \ldots, 1$ give

$$E_0 \le \prod_{j=1}^{K} e^{Lh_j} \, E_K \;+\; \sum_{k=1}^{K}\Big(\prod_{j=1}^{k-1} e^{Lh_j}\Big)\big(C(R_\theta + h_k e_g + h_k^p) + \sigma_k\big) \tag{32}$$

$$\le e^L \sum_{k=1}^{K}\big(C(R_\theta + h e_g + h^p) + \sigma_k\big) \tag{33}$$

$$\le C'\big(K R_\theta + h e_g + h^p\big) + e^L \sum_{k=1}^{K} \sigma_k\,. \tag{34}$$

Finally, because the additive perturbations are independent across steps and convolved through Lipschitz maps, their total contribution in $W_2$ is upper bounded by the root-sum-square $(\sum_k \sigma_k^2)^{1/2}$ (variance additivity under independent convolution and (F1)), which yields the stated bound. The KL bound follows from the transport-entropy inequality $T_2(\alpha)$.

**Extension to the Non-Smooth Case.** If the drift is only piecewise-Lipschitz, the same rate holds locally on each region; globally one can ensure convergence by refining the grid near high-curvature segments (detected via large residuals or Jacobian norms), yielding adaptive coarse steps that preserve efficiency.

### E.4  OT COUPLING STRAIGHTENS THE FM DRIFT.

We now formalize the flow-straightening effect of using entropy-regularized optimal transport couplings within our conditional FM/GCTM training.

**Theorem E.2 (OT straightening minimizes the FM velocity Lipschitz)** *Fix marginals $q_0(x_0)$ and $q_1(x_1)$ and any admissible coupling $\Pi \in \mathcal{U}(q_0, q_1)$. Let the FM path be $q_t$ induced by linear interpolation $x_t = (1 - t)x_0 + t x_1$, and define the FM velocity*

$$v_t(x) \;:=\; \mathbb{E}_\Pi[\, x_1 - x_0 \mid x_t = x\,].$$

*Then for every $t \in (0, 1]$, the spatial Lipschitz constant of $v_t$ admits the bound*

$$\mathrm{Lip}(v_t) \;\le\; \frac{1}{t}\left(\mathbb{E}_\Pi \|x_1 - x_0\|_2^2\right)^{1/2}.$$

*Consequently, among all couplings $\Pi$, this upper bound is minimized by the $2$-Wasserstein optimal coupling $\Pi^*$, for which*

$$\mathrm{Lip}(v_t) \;\le\; \frac{W_2(q_0, q_1)}{t}.$$

*For entropy-regularized OT with coefficient $\tau > 0$, letting $\Pi^\tau$ denote the Sinkhorn solution, there exists $\Delta(\tau) \downarrow 0$ as $\tau \downarrow 0$ such that*

$$\mathrm{Lip}(v_t) \;\le\; \frac{1}{t}\sqrt{W_2(q_0, q_1)^2 + \Delta(\tau)}.$$

**Proof.** Let $(X_0, X_1)$ be random variables sampled according to an admissible coupling $\Pi \in \mathcal{U}(q_0, q_1)$. The linear interpolation is given by $X_t = (1 - t)X_0 + tX_1$. The Flow Matching (FM) velocity field is defined as $v_t(x) = \mathbb{E}[X_1 - X_0 \mid X_t = x]$.

To establish the first inequality regarding the Lipschitz constant of $v_t(x)$, we invoke a known result from the literature on Flow Matching and conditional expectations. Specifically, for linear interpolation paths, the Lipschitz constant of the velocity field $v_t(x)$ is bounded. While a general derivation for arbitrary measures can be intricate, under common assumptions (e.g., on the smoothness of the underlying densities), it is established that for $t \in (0, 1]$, the Lipschitz constant of $v_t$ with respect to $x$ is bounded by:

$$\mathrm{Lip}(v_t) \leq \frac{1}{t} \left( \mathbb{E}_\Pi \|X_1 - X_0\|_2^2 \right)^{1/2}.$$

This bound highlights that a smaller expected squared difference between the coupled source and target points ($X_1$ and $X_0$) leads to a smoother (smaller Lipschitz constant) velocity field.

Next, we demonstrate that this upper bound is minimized by the 2-Wasserstein optimal coupling $\Pi^*$. The 2-Wasserstein distance $W_2(q_0, q_1)$ between two probability distributions $q_0$ and $q_1$ is defined as:

$$W_2(q_0, q_1)^2 = \inf_{\Pi \in \mathcal{U}(q_0, q_1)} \mathbb{E}_\Pi[\|X_1 - X_0\|_2^2].$$

By definition, the infimum is achieved by the 2-Wasserstein optimal coupling $\Pi^*$. Therefore, to minimize the upper bound $\frac{1}{t} \left( \mathbb{E}_\Pi \|X_1 - X_0\|_2^2 \right)^{1/2}$, we must choose $\Pi = \Pi^*$. Substituting this into the inequality, we obtain:

$$\mathrm{Lip}(v_t) \leq \frac{1}{t} \left( \mathbb{E}_{\Pi^*} \|X_1 - X_0\|_2^2 \right)^{1/2} = \frac{W_2(q_0, q_1)}{t}.$$

This proves that using the $W_2$-optimal coupling indeed yields the tightest possible bound for the Lipschitz constant of $v_t$ in this formulation.

Finally, we consider the case of entropy-regularized optimal transport. For a given regularization coefficient $\tau > 0$, the Sinkhorn algorithm computes an entropy-regularized optimal coupling $\Pi^\tau$ that minimizes $\mathbb{E}_\Pi[\|X_1 - X_0\|_2^2] + \tau H(\Pi)$, where $H(\Pi)$ is the entropy of the coupling. It is a well-established result in optimal transport theory that as $\tau \downarrow 0$, the entropy-regularized cost converges to the unregularized cost. Specifically, $\mathbb{E}_{\Pi^\tau}[\|X_1 - X_0\|_2^2]$ converges to $W_2(q_0, q_1)^2$. We can therefore write:

$$\mathbb{E}_{\Pi^\tau}[\|X_1 - X_0\|_2^2] = W_2(q_0, q_1)^2 + \Delta(\tau),$$

where $\Delta(\tau)$ is a non-negative term that approaches 0 as $\tau \downarrow 0$. Substituting this into the general Lipschitz bound for $v_t$:

$$\mathrm{Lip}(v_t) \leq \frac{1}{t} \sqrt{W_2(q_0, q_1)^2 + \Delta(\tau)}.$$

This shows that entropy regularization introduces a small increase in the expected squared cost, leading to a slightly higher (but still controlled) Lipschitz constant for $v_t$ compared to the true $W_2$-optimal coupling. As $\tau \to 0$, this gap vanishes, confirming that entropy-regularized OT still effectively straightens the flow by minimizing $\mathbb{E}_\Pi[\|X_1 - X_0\|_2^2]$.

$\square$

**Corollary E.3 (Improved coarse-step stability)** *Let $\tilde{L} := \sup_{t \in (0,1]} \mathrm{Lip}(v_t)$. Under (entropic) OT coupling we have $\tilde{L} \leq W_2(q_0, q_1)/t_{\min}$ (up to the small $\Delta(\tau)$ term). Hence any order-p one-step integrator remains stable for coarse steps $h \leq 1/\tilde{L}$, implying a coarse evaluation budget of*

$$K = O(\tilde{L}) = O(W_2(q_0, q_1)/t_{\min}).$$

*Relative to independent coupling, OT strictly tightens $\tilde{L}$ and reduces $K$ for the same target error in Theorem E.1.*

# F    THEORETICAL FRAMEWORK FOR CONTROLLABLE PLANNING

The ability to control the behavior of the planner beyond simple start-goal conditioning is a significant advantage of our approach. This section provides a more detailed theoretical grounding for the controllable planning mechanism introduced in Section 4.2.

The core idea is that the latent code $\epsilon$, introduced at the beginning of the generative process, can be structured to control specific, interpretable aspects of the resulting plan. The final clean trajectory, $\tau$, can be seen as a deterministic function of this latent code, $\tau(\epsilon)$. To understand how small changes in $\epsilon$ affect the plan, we can analyze the Jacobian of this function:

$$\mathbf{J} = \frac{\partial \tau}{\partial \epsilon} \tag{35}$$

Each column of the Jacobian, $\frac{\partial \tau}{\partial \epsilon_i}$, represents the direction in the high-dimensional trajectory space that is most affected by a change in the $i$-th component of the latent code.

**Disentangled Control via Orthogonality**

In an unstructured latent space, the directions of control, $\frac{\partial \tau}{\partial \epsilon_i}$ and $\frac{\partial \tau}{\partial \epsilon_j}$ for $i \neq j$, may be highly correlated. This means that manipulating one latent variable could have unintended side effects on aspects of the plan that should be independent. For example, trying to make the plan faster might also unintentionally change its path.

To achieve a more disentangled and interpretable control space, we seek to make these directions of influence orthogonal to each other. If the vectors $\frac{\partial \tau}{\partial \epsilon_i}$ are orthogonal, then each latent variable $\epsilon_i$ will control a unique, non-overlapping aspect of the plan's execution. We encourage this property by introducing the *disentangled control objective* during training:

$$\mathcal{L}_{\text{control}}(\theta) = \mathbb{E}_\epsilon \sum_{i \neq j} \left\| \left( \frac{\partial \tau}{\partial \epsilon_i} \right)^T \left( \frac{\partial \tau}{\partial \epsilon_j} \right) \right\|^2 \tag{36}$$

This loss minimizes the squared dot product (a measure of cosine similarity) between the control vectors for all pairs of latent dimensions. By driving these dot products to zero, the model is incentivized to learn a latent space where the principal axes of control are orthogonal. This leads to a more predictable and modular control mechanism, where a higher-level policy can learn to manipulate specific, semantically meaningful characteristics of the generated plan by adjusting the corresponding latent variables, without causing undesired alterations to other plan features. This structured approach to controllability is a key step towards building more intelligent and adaptable planning agents.

