# OpenReview forum: "MCPlanner: Multi-Scale Consistency Planning for Offline Reinforcement Learning"
_ICLR.cc/2026/Conference — Submitted to ICLR 2026_

### Official Review · Reviewer_yiK8 · 2025-10-31

**Soundness:** 3
**Presentation:** 3
**Contribution:** 3
**Rating:** 6
**Confidence:** 4

**Summary:**

This paper introduces MCPlanner, a multi-scale planning framework for offline goal-conditioned reinforcement learning (RL). The idea is to replace the traditional separated hierarchical structure (high-level and low-level polices) with a single unified model based on the Generalized Consistency Trajectory Model (GCTM). Comprehensive experiments on the OGBench benchmark show that MCPlanner consistently outperforms state-of-the-art diffusion-based and hierarchical planners.

**Strengths:**

* The paper presents a conceptually novel approach to long-horizon planning in offline RL by unifying hierarchical diffusion planning into a single flow-based model (GCTM). The introduction of a multi-scale consistency loss and conditional optimal-transport coupling represents a creative and principled extension of flow-matching ideas to hierarchical planning.

* The experiments are extensive across 35 tasks in the OGBench benchmark, including both locomotion and manipulation domains. Empirical results consistently demonstrate superior performance over previous hierarchical diffusion methods.

**Weaknesses:**

### 1. Clarity and Accessibility :

The paper is not easy to follow, particularly in the methodological exposition. Some sections could benefit from clearer explanations and more intuitive motivation. For instance, Figure 2 is difficult to interpret and lacks sufficient textual explanation in the main body. The authors should explicitly describe how querying the same GCTM across different time scales results in the emergence of hierarchical behavior. Furthermore, the discussion about why separately trained high-level and low-level planners lead to lack of coherence and consistency remains underdeveloped. Providing a more concrete analysis (or illustrative example) of how the two-model approach fails to maintain consistency would greatly improve readability and conceptual clarity.

### 2. Unclear causal link between performance and claimed coherence :

Although the experiments show that MCPlanner achieves better performance than baselines, and that removing  L_MS leads to degradation,
it is not clearly demonstrated that these gains directly stem from improved coherence or consistency. Additional analysis, such as qualitative visualizations of subgoal alignment or temporal consistency metrics, would strengthen the causal argument between the multi-scale objective and the resulting performance.

### 3. Inference-time advantage and fairness concerns :

While the paper reports consistent superiority over diffusion-based baselines, MCPlanner’s inference pipeline includes candidate generation, scoring, and latent refinement, effectively adding test-time optimization and selection steps. These additional computations likely improve success rates but were not applied to baseline methods, making the comparison potentially unfair in terms of compute and inference strategy.
Indeed, Figure 3 shows that when the number of trajectory candidates is reduced to N=1, MCPlanner’s performance drops substantially, approaching that of baseline planners. This suggests that part of the reported gain may come from the inference-time candidate search rather than the proposed architecture itself.

### 4. Lack of quantitative analysis on computational efficiency :

The paper claims that the unified GCTM design reduces training and inference cost compared to two-model hierarchies, yet provides no quantitative evidence, such as runtime, GPU-hour, or FLOPs per trajectory. Given that GCTM integration and OT coupling introduce additional computation (e.g., Sinkhorn iterations, multi-time evaluations), a fair and transparent assessment of computational cost is needed to justify the efficiency claims.

**Questions:**

Please provide your response to the weaknesses mentioned above.

---

### Official Review · Reviewer_e2iD · 2025-10-31

**Soundness:** 2
**Presentation:** 2
**Contribution:** 2
**Rating:** 2
**Confidence:** 4

**Summary:**

This paper addresses long-horizon planning in offline reinforcement learning, focusing on multi-scale planning. The authors propose a multi-scale planner built upon Generalized Consistency Trajectory Models (GCTM). Unlike hierarchical planners that require separate models across temporal scales, this method integrates all resolutions into a single unified model, aiming for improved efficiency and coherence.

**Strengths:**

1. The motivation for tackling multi-scale planning in offline RL is clearly presented and well-justified, connecting the need for temporal abstraction with the limitations of current planners.

2. The paper is easy to follow, self-contained, and includes well-written preliminaries that make it accessible to a broad ICLR audience.

**Weaknesses:**

1. The motivation for using GCTM as the specific backbone is insufficiently justified. While GCTMs are relevant to consistency-based generative modeling, the paper does not explain why they are better suited for multi-scale planning than diffusion-based planning.

2. The comparative evaluation lacks a controlled ablation isolating the effect of the proposed multi-scale mechanism. Current results compare against planners that differ in both backbone and architecture, making it unclear whether the gains stem from multi-scale or model-specific characteristics. As a suggestion, authors can include a GCTM variant without the multi-scale mechanism.

3. The novelty is somewhat limited. Multi-scale temporal modeling has been previously explored in diffusion-based planners ([1], [2]). The contribution thus appears incremental, extending known ideas to a new architecture (GCTM) without clear conceptual gain.

**Questions:**

1. The text under the images in Figure 2 is ambiguous. At first glance, it seems to correspond to textual input, but the method is not applied to text-based environments. Could you clarify what this text represents?

2. In Table 1, the reported score for cube-quadruple-play-v0 is zero across all baselines. Could the authors elaborate the reason?

3. The authors attribute the gap between two-level hierarchy and the proposed multi-scale variant to "coherence gap and compounding errors". Could you provide quantitative or qualitative evidence supporting this claim?

---

### Official Review · Reviewer_Wbtz · 2025-11-01

**Soundness:** 2
**Presentation:** 3
**Contribution:** 2
**Rating:** 4
**Confidence:** 2

**Summary:**

This paper proposes MCPlanner, a unified planning framework built on Generalized Consistency Trajectory Models (GCTMs) that replaces the usual two-model hierarchy with a single model queried at multiple temporal resolutions. Specifically, this work proposes: (i) a multi-scale consistency objective to align long jumps with compositions of short jumps; and (ii) conditional optimal-transport coupling to straighten the learned flow and stabilize coarse integration.

**Strengths:**

The proposed method has the advantages of generating (1) coarse intermediate “*subgoals*” very quickly; (2) fine-grained motion on interval level.

This work proposes a new entropy regularized OT coupling between batches of expert trajectories and their corresponding deterministic trajectory priors to reduce the curvature of the FM ODE.

Ablation study demonstrated the effectiveness of the design choice of the proposed method.

**Weaknesses:**

Although this paper proposed some modifications over GCTM, the overall technical novelty and contribution in terms of consistency models are just incremental as the key technique used is based on existing training objectives.

The central topic of the paper is more on goal achieving in offline planning tasks like navigation or locomotion. There is no reward function used in the formulation or the methodology. So this paper is strongly related to offline RL where offline policy improvement is the key objective. Therefore, the paper is closer to trajectory planning than offline RL.

It would be better to compare with non-diffusion based methods like hierarchical RL.

There are some confusions over the notations, for example, T is used for trajectory length and is also used for diffusion/FW denoising steps.

It seems that the method cannot handle test-time environmental uncertainty and errors as the planning seems to be done before execution. Can the proposed method plan in close loop in a receding horizon fashion?

**Questions:**

Please see the weaknesses part.

---

### Meta-Review · Area_Chair_4QSt · 2025-12-29

**Summary:**

This paper introduces Multi-scale Consistency Planner (MCPlanner), a framework that leverages the unique properties of Generalized Consistency Trajectory Models (GCTMs) to create a fluid and unified planning hierarchy. The idea is to replace the traditional separated hierarchical structure (high-level and low-level polices) with a single unified model based on the GCTM.

**Reviewer Concerns:**

The reviewers raised a few major concerns: 1) One common concern by all three reviewers is with the novelty, since the key techniques used in this paper have been previously explored in diffusion-based planners (e.g., [1], [2]), so the overall technical contributions in terms of consistency models are incremental. 2) The comparative evaluation against baselines is inadequate: One reviewer pointed out that MCPlanner’s inference pipeline includes candidate generation, scoring, and latent refinement, effectively adding test-time optimization and selection steps. These additional computations likely improve success rates but were not applied to baseline methods, making the comparison potentially unfair in terms of compute and inference strategy. Another reviewer raised the concern that the comparative evaluation lacks a controlled ablation isolating the effect of the proposed multi-scale mechanism.

The authors did not provide any rebuttal.

**Reviewer Scores:**

The scores are 4/2/6.

---

### Decision · Program_Chairs · 2026-01-26

Reject